# Volatility Spillovers among Developed and Developing Countries: The Global Foreign Exchange Markets

**Walid Abass Mohammed**

College of Business, Technology and Engineering, Sheffield Hallam University, Sheffield S1 1WB, UK; w.mohammed@shu.ac.uk

**Abstract:** In this paper, we investigate the "static and dynamic" return and volatility spillovers' transmission across developed and developing countries. Quoted against the US dollar, we study twenty-three global currencies over the time period 2005–2016. Focusing on the spillover index methodology, the generalised VAR framework is employed. Our findings indicate no evidence of bi-directional return and volatility spillovers between developed and developing countries. However, unidirectional volatility spillovers from developed to developing countries are highlighted. Furthermore, our findings document significant bi-directional volatility spillovers within the European region (Eurozone and non-Eurozone currencies) with the British pound sterling (GBP) and the Euro (EUR) as the most significant transmitters of volatility. The findings reiterate the prominence of volatility spillovers to financial regulators.

**Keywords:** foreign exchange market; volatility spillover; return spillover; VAR framework; variance decomposition; financial crisis; financial interdependence

## 1. Introduction

The increasing financial interdependence, particularly during the current era of global economic events and financial turbulence has prompted considerable interest from market participants and academic research. While much attention has been paid to the magnitude of return and volatility spillovers across global stock markets, little is known about the foreign exchange channels, in particular, the foreign exchange markets' channels between developed and developing countries. Some studies of this kind have investigated the return co-movements and volatility spillovers, primarily across the developed countries Anderson et al. (2001), Pérez-Rodrìguez (2006), Boero et al. (2011), and Rajhans and Jain (2015). Others have considered the regional spillover transmission and produced insignificant results.

However, given the trillions of dollars of exchange rate trading in international financial markets, it is important to fully understand and investigate in greater depth the potential spillovers of international currencies. This is an important aspect that investors must account for during the formation of their position and portfolios. Before the recent financial turmoil, the contribution of the channels of foreign exchange markets' spillovers to the global financial instability, for some, appeared to be less worrisome. Whereas in fact, the behaviour of the stock prices (which have been extensively studied) is mainly explained by volatilities in the foreign exchange markets Kim (2003).

Thus, in this paper, we provide new insights into the incomplete investigation of the channels of global intra-foreign exchange markets' spillovers. Our key question is whether the effect of return and volatility spillovers is bidirectional between developed and developing countries. This is because of the recent financial crisis which originated

in major financial hubs in developed countries, primarily in the USA, that developing countries are not responsible for, nevertheless, are seriously affected by it. To study the return and volatility spillovers' transmission, we model the daily spot exchange rates for 23 global currencies. According to the BIS (2013), the USD, EUR, GBP, AUD, CAD, JPY, and CHF are the most traded globally and account for almost 90 per cent of the global foreign exchange turnover. In particular, we adopt the generalised vector autoregressive (VAR) approach focusing on the variance decomposition of Diebold and Yilmaz (2009).

The innovative feature of this approach, in addition to being rigorous, is that it allows the aggregation of valuable information across markets into a single spillover index. The unique structure of the spillover index is designed to unleash an in-depth analysis of the negative spillovers' transmission across markets, i.e., how a shock in a particular market is due to exogenous/endogenous shocks to other markets.

Financial crises are almost difficult to predict; nevertheless, it is important to identify fluctuations in volatility over a different time period. Thus, we examine the time-varying net volatility spillovers using the autoregressive conditional heteroskedasticity (ARCH) model. The time-varying volatility identifies the specific points of significant shifts in volatility spillovers during the time period of our sample (2005–2016). We provide evidence of significant volatility clustering during the 2008 financial crisis. The ARCH model, which was first introduced by Engle (1982) is widely used in the literature Bollerslev et al. (1994), Kaur (2004), Basher et al. (2007) for its ability to capture persistence in time-varying volatility based on squared returns. Most importantly, to investigate the nature of the net volatility and net pairwise spillover effects, we implement Diebold and Yilmaz (2012) methodology. By doing so, we are able to show the difference between the amount of the gross volatility shocks within our sample that were transmitted to and received from developed and developing countries. To enhance the reliability of the findings, we provide evidence in different dimensions (using a sample of twenty-three global currencies over the time period 2005–2016). The first is the static analysis approach, which provides results in the form of spillover tables. The second is the dynamic analysis, which yields the spillover plots. Third, is the time-varying net volatility results, which we provide in the form of figures. Finally, the net volatility and net pairwise spillover effects.

The analysis is based on a large daily spot exchange rates' dataset, which covers a long period pre and post the most recent events in the global economy. In particular, in this paper, we provide results based on extensive empirical analyses, such as the spillover index (both static and dynamic analyses), time-varying net volatility, net volatility, and net pairwise volatility effects.

Guided by the empirical approach described above, the main findings indicate that there is no evidence of bidirectional volatility spillovers between developed and developing countries. Although unsurprisingly, the results highlight evidence of unidirectional volatility spillovers pouring from developed to developing countries. In particular, the volatility spillovers from developed to developing countries seemed to be specifically strong following the collapse of Lehman Brothers in 2008. Another curious outcome of the findings is that developed countries are the highest receivers and transmitters of volatility spillovers, dominated by the British pound sterling, the Australian dollar, and the Euro, whereas developing countries are net receivers of volatility spillovers. The findings, therefore, indicate that the currency crisis tends to be regional Glick and Rose (1998) and Yarovaya et al. (2016).

Meanwhile, in light of the recent financial crisis, the analytical results demonstrate that the cross-country spillover activities between developed and developing countries are insignificant. Although the financial risk propagated during the 2008 financial crisis engulfed the global economy, that being said, because of the recent financial markets' developments, such as financial engineering (collateral debt obligation, credit default swap, and derivative securities), financial risks triggered different means of spreading across the global economy, which still need to be discovered, understood, and described appropriately.

The remainder of this paper is organised as follows: In Section 2, we discuss some critical arguments presented in the related literature; in Sections 3 and 4, we introduce the data used in the analysis and the empirical methodology applied; in Section 5, we provide empirical results, including the robustness and some descriptive statistics; in Section 6, we discuss the time-varying volatility; in Section 7, we introduce the net spillovers and net pairwise volatility spillovers; in Section 8, we provide conclusions; and in Section 9, we discuss the study limitations and future research.

## 2. Related Literature

This brief review of the literature is focused on the foreign exchange markets' spillover channels, which is one of the most intensely debated issues in recent literature. However, the significance of the foreign exchange markets' spillover channels to the stability of financial markets was acknowledged three decades ago. For example, Engle et al. (1988) established the first thread-tying efforts of the intra-day exchange rate's volatility spillovers within one country (heat waves) and across borders (meteor shower). The "heat waves" is a hypothesis that indicates that volatility in one market may continue in the same market the next day, whereas the "meteor shower" is a phenomenon, which implies that volatility in one market may spillover to another market.

In this paper, the authors provide evidence of transmitted volatility spillover from one market to another. This opening up, particularly after the 2008 financial crisis, amplified the importance of the spillover channels in the stock and the foreign exchange markets. This is due to the repercussions of the shocking types of financial risks stemming from the interconnected nature of the financial markets. Thus, there is a shred of growing evidence in the literature that supports the association of return and volatility spillovers with global economic events and financial crises (for reviews see Diebold and Yilmaz (2009), Beirne et al. (2013), Yilmaz (2009), Gebka (2012), Jung and Maderitsch (2014), Ghosh (2014), Choudhry and Jayasekera (2014), Antonakakis et al. (2015), Mozumder et al. (2015)). In addition, the recent financial crisis demonstrates the severity of the cross-market volatility spillovers, which transmitted across countries through the stock and foreign exchange markets' channels (for reviews see Fedorova and Saleem (2009), Cecchetti et al. (2011), Awartani and Maghyereh (2013), Jouini (2013), Shinagawa (2014), Do et al. (2015)). Another important feature of the foreign exchange spillover channels is that their effects act differently during, before, and after the economic events and financial crisis episodes. For example, based on VAR models, Diebold and Yilmaz (2009) examined nineteen global equity markets from the 1990s to 2009. They found striking evidence that return spillovers displayed a slightly increasing trend but no bursts, while volatility spillovers displayed no trend but strong bursts associated with crisis events.

In addition, the effect of return and volatility spillovers may extend to the business cycle mechanism. Several studies Imbs (2004), Eickmeier (2007), Imbs (2010), and Claessens et al. (2012) have argued that volatility spillovers inflicted business cycle synchronisation between countries through different channels. These channels mainly include the exchange rate, confidence, trade, and financial integration channel. Antonakakis et al. (2015) suggested that the spillover effect could also be transmitted through business cycles across countries. According to Eickmeier (2007), the confidence channel represented the response from domestic agents to the potential spillovers coming from foreign shocks to the local economy.

Several studies have attempted to empirically analyse the exchange rate co-movements and volatility spillovers across countries, in particular, the financial transmission between the Euro (EUR), British pound sterling (GBP), Australian dollar (AUD), Swiss franc (CHF), and the Japanese yen vis-à-vis the US dollar. For instance, Boero et al. (2011) and Rajhans and Jain (2015) found a high correlation between the Euro and the British pound sterling against the US dollar and that the British pound sterling was a net receiver. Nikkinen et al. (2006) studied the future expected volatility linkages among major European currencies (the Euro, the British pound sterling, and the Swiss franc) against the US

dollar. They found future volatility linkages between the major currencies and that the British pound sterling and the Swiss franc were significantly affected by the implied volatility of the Euro. Boero et al. (2011) found an increase in co-movements between the Euro and the British pound sterling after the introduction of the Euro as compared with the pre-euro era. A different perspective was offered by Antonakakis (2012), who used the VAR model, and found significant return co-movements and volatility spillover between major exchange rates before the introduction of the Euro, which lowered during the post-euro periods.

Baruník et al. (2017) analysed asymmetries in volatility spillovers on the foreign exchange market. Applying high-frequency data of the most actively traded currencies over the time period 2007–2015, the authors found that negative spillovers dominated positive spillovers. Katusiime (2019) evaluated the spillover effects between foreign exchanges and found conditional volatility among currency rates and commodity price in Uganda. Most recently, Fasanya et al. (2020) investigated the dynamic spillovers and connectedness between the COVID-19 pandemic and global foreign exchange markets. They found a high degree of interdependence between the global COVID-19 occurrences and the returns' volatility of the majorly traded currency pairs.

In this paper, we have based our measurements of return and volatility spillovers on vector autoregressive (VAR) models, which have been laid out in recent studies Palanska (2018), Mensi et al. (2018), and Fasanya et al. (2020). As compared with other methodologies such as MGARCH and the realised volatility (RV) estimator, the VAR provides strikingly accurate results using high-frequency data. This is because the realised volatility (RV) estimator is considered to be biased at a high-frequency sampling Barndorff-Nielsen et al. (2009) and Floros et al. (2020).

The previously discussed papers established the evidence of return co-movements and volatility spillovers across developed countries' exchange rates. It is also important to examine the behaviour of asset return and volatility spillovers of the foreign exchange markets between developed and developing countries. Notwithstanding, only a few of the studies (which focused mainly on central Europe's foreign exchange markets) have produced limited results. For example, using a multivariate GARCH model, Lee (2010) studied volatility transmission across ten emerging foreign exchange markets and provided evidence of regional and cross countries' volatility spillovers. Bubák et al. (2011) examined the volatility transmission across three central European's emerging markets, in particular, among Czech, Hungarian, and Polish currencies. Their main finding was significant intra-regional volatility spillovers across central Europe's foreign exchange markets.

In contrast to the above-mentioned studies, in this paper, we provide a thorough investigation of return and volatility spillovers between developed and developing countries, in particular, the transmission through foreign exchange markets' channels. We examine broad data samples from twenty-three developed and developing countries (which have received somewhat limited attention) before, during, and after the 2008 financial crisis. The extended data sample from 2005 to 2016 emphatically help, in a way, to unfold the effect of return and volatility spillovers across global foreign exchange markets, which currently dominate the focus of policymakers as well as financial managers.

## 3. Database and Methodology

### 3.1. Database

In this paper, the underlying data consist of daily spot exchange rates of currencies, comprising a total of twenty-three developed and developing countries across the world vis-à-vis the US dollar. Taken from DataStream Thomson Reuters through the WM/Reuters channel, the sample period starts on 31 May 2005 and ends on 1 June 2016. Our study period facilitates the production of comprehensive and precise measures of return spillo-

vers and volatility spillovers pre and post the 2008 financial crisis. The series under investigation included currencies from eleven developed countries (i.e., the British pound sterling (GBP), the Euro (EUR), the Australian dollar (AUD), the Canadian dollar (CAD), the Swiss franc (CHF), the Japanese yen (JPY), the Icelandic krona (ISK), the Czech Republic koruna (CZK), the Hong Kong dollar (HKD), the Singapore dollar (SGD), and the South Korean won (KRW)) and currencies from twelve developing countries (i.e., the Russian ruble (RUB), the Turkish lira (TRY), the Indian rupee (INR), the Indonesian rupiah (IDR), the Argentine peso (ARS), the Malaysian ringgit (MYR), the Thai baht (THB), the Mexican peso (MXN), the Saudi Arabian riyal (SAR), the United Arab Emirates dirham (AED), the South African rand (ZAR), and the Nigerian naira (NGN)). According to the Bank for International Settlement (BIS) report (2013), our underlying chosen samples include the most actively traded currencies across the financial markets globally.

### 3.2. Obtaining Daily Returns

To obtain the daily returns series, we calculate the daily change in the log price of close data. When price data are not available for a given day due to a holiday or in the case of omitted value, we use the previous day value. As spot rates are non-stationary, we calculate the daily returns as:

$$r_t = ln(y_t) - ln(y_{t-1}),$$

where $y_t$ is the spot exchange rate at time *t*, with *t* = 1, 2, ..., T, and the natural logarithm *ln*. Table 1 provides a variety of descriptive statistics for returns.

### 3.3. Obtaining Daily Return Volatilities

A different approach could be employed to achieve the global foreign exchange market historical volatility. However, in this paper, we employ the improved estimators of security price fluctuations of Garman and Klass (1980) and Alizadeh et al. (2002). The instinct of this methodology is that the underlying volatility estimators are based on historical opening, closing, high and low prices, and transaction volume. The underlying model assumption is that the diffusion process governs security prices:

$$P(t) = \emptyset\big(B(t)\big) \tag{1}$$

where $P$ represents the security price, $t$ is time, $\emptyset$ is a monotonic time-independent transformation, and $B \langle t \rangle$ is a diffusion process with differential representation. The use of monotonicity and time independence both assure that the same set of sample paths generate the sample maximum and minimum values of $B$ and $P$ Garman and Klass (1980):

$$dB = \sigma \, dz \tag{2}$$

where $dz$ is the standard Gauss–Wiener process and $\sigma$ is an unknown constant to be estimated. Implicitly, the phenomenon is dealing with the transformed "price" series, the geometrical price would mean logarithm of the original price and volatility would mean "variance" of the original logarithmic prices. The original root of the Garman and Klass methodology is the Brownian motion, where they added three different estimation methods. They based their methodology estimation on the notion of historical opening, closing, high and low prices, and the transaction volume, through which they provided the following best analytic scale-invariant estimator:

$$\sigma_t = \sqrt{\frac{N}{n} \cdot \sum_{i=1}^{N} \frac{1}{2} \cdot (\log\left(\frac{H_i}{L_i}\right))^2 - (2.\log(2) - 1). \, \log{(\frac{C_i}{O_i})^2}} \tag{3}$$

where $\sigma_t$ is an unknown constant to be estimated, $N$ is the number of trading days in the year, $n$ is the chosen sample, $H$ is today's high, $L$ is today's low, $O$ and $C$ are today's opening and closing, respectively. Explaining the coefficients of the above formulae is beyond the scope of this paper for now. However, to obtain the foreign exchange market's

volatilities, we use intra-day high, low, opening, and closing data. When price data are not available for a given day due to a holiday or in the case of omitted value, we use the previous day value. Table 2 shows descriptive statistics for the global foreign exchange market's volatilities.

## 4. Methodology

To examine return and volatility spillovers across our sample, we use the generalised vector autoregressive (VAR) methodology. In particular, our investigation is based on the variance decompositions proposed by Diebold and Yilmaz (2009). The concept of variance decomposition is very rigorous and helpful as it allows the aggregation of valuable information across markets into a single spillover index. In other words, how shocks in market A are due to exogenous shocks to other markets. We can express this phenomenon through variance decomposition concomitant with an N-variable VAR by adding the shares of the forecast error variance for each asset $i$ coming from shocks to an asset $j$, for all $j \neq i$ tallying up across all $i = 1, \dots,$ N. Then, considering the example of simple covariance stationary first-order two-variable VAR,

$$x_t = \Phi x_{t-1} + \varepsilon_t \tag{4}$$

where $x_t = (x_{1t,} x_{2t})$ and $\Phi$ is a parameter matrix. In the following empirical work, $x$ will be either a vector of foreign exchange returns or a vector of foreign exchange return volatilities. The moving average representation of the VAR is given by:

$$x_t = \Theta(L)\varepsilon_t \tag{5}$$

where $\Theta(L) = (1 - \Phi L)^{-1}$ which for simplicity could be rewritten as:

$$x_t = A(L)\, u_t \tag{6}$$

where $A(L) = \Theta(L)Q^{-1}$, $u_t = Q_t\, \varepsilon_t$, $E(u_t\, u') = 1$, and $Q^{-1}$ is the unique Cholesky factorisation of the covariance matrix of $\varepsilon_t$. Then, considering the one-step-ahead forecast, the precise approach would be the Wiener–Kolmogorov linear least-squares forecast as:

$$x_t + 1, t = \Phi x_t \tag{7}$$

With corresponding one-step-ahead error vector:

$$e_t + 1, t = x_{t+1} - x_{t+1,t} = A_0\, u_{t+1} = \begin{bmatrix} a_{0,11} & a_{0,12} \\ a_{o,21} & a_{0,22} \end{bmatrix} \begin{bmatrix} u_{1,t+1} \\ u_{2,t+1} \end{bmatrix} \tag{8}$$

And comprises the following covariance matrix:

$$E(e_{t,+1,t}\, e'_{+1,t}) = A_0 A'_0. \tag{9}$$

To clarify, the variance of the one-step-ahead error in forecasting $x_{1t}$ is $a_{0,11}^2 + a_{0,12}^2$ and the variance of the one-step-ahead error in forecasting $x_{2t}$ is $a_{0,21}^2 + a_{0,22}^2$. Diebold and Yilmaz (2009) utilised the mechanism of variance decompositions to split the forecast error variances of each variable into parts attributable to a broader system shock. This facilitates answering the question of what fraction of the one-step-ahead error variance in forecasting $x_1$ is due to shocks to $x_1$ and shocks to $x_2$. Likewise, what portion of the one-step-ahead error variance in forecasting $x_2$ is due to shocks to $x_1$ and shocks to $x_2$.

### 4.1. The Spillover Index

The spillover index of Diebold and Yilmaz (2009) represents the fractions of the one-step-ahead error variances in forecasting $x_i$ due to shocks to $x_j$, for $i, j = 1, 2, i \neq j$. These two variables construct the spillover index with two possible spillovers' outcomes. First, $x_{1t}$ represents shocks that affect the forecast error variance of $x_{2t}$ with the contribution $(a_{0,21}^2)$. Second, $x_{2t}$ similarly represents shocks that affect the forecast error variance of

$x_{1t}$ with a contribution of $(a_{0,12}^2)$ totaling the spillover to $a_{0,12}^2 + a_{0,21}^2$. This can be expressed relative to the total forecast error variation as a ratio percentage projecting the spillover index as:

$$s = \frac{a_{0,12}^2 + a_{0,21}^2}{trace(A_0 A'_0)} \times 100 \qquad (10)$$

The spillover index can be sufficiently generalised to wider dynamic environments, particularly for the general case of a $p^{th}$-order N-variable VAR, using H-step-ahead forecast as:

$$s = \frac{\sum_{h-0}^{H-1} \sum_{\substack{i,j=1 \\ i \neq j}}^{N} a_{h,ij}^2}{\sum_{h=o}^{H-1} trace(A_h A'_h)} \times 100 \qquad (11)$$

To examine the data, the spillover index described above allows the aggregation degree of cross-market spillovers across the large data, which consist of 2872 samples into a single spillover measure. We use second-order 23 variables with 10-step-ahead forecasts. Tables 3 and 4 show the results of return spillover and volatility spillover, respectively.

*4.2. Net Spillovers*

To generate the net volatility spillovers, we follow (Diebold and Yilmaz 2012) by first calculating the directional spillovers, which can be achieved by normalising the elements of the generalised variance decomposition matrix. This way, we can measure the directional volatility spillovers received by (developing) countries from the developed countries or vice versa as follows:

$$S_{i.}^{\text{g}} = \frac{\sum_{\substack{j=1 \\ j \neq i}}^{N} \tilde{\theta}_{ij}^{\text{g}}(H)}{\sum_{i,j=1}^{N} \tilde{\theta}_{ij}^{\text{g}}(H)} . 100 = \frac{\sum_{\substack{j=1 \\ j \neq i}}^{N} \tilde{\theta}_{ij}^{\text{g}}(H)}{N} . 100. \qquad (12)$$

Thus, from the above equation, the net volatility spillovers can be obtained from market *i* to all other markets *j* as follows:

$$S_i^{\text{g}}(H) = S_{.i}^{\text{g}} - S_{.i}^{\text{g}}(H). \qquad (13)$$

*4.3. Net Pairwise Spillovers*

Given the net volatility spillovers described in Equation (12), which provides the net volatility of each market contribution to others, then, it is relatively easy to examine the net pairwise volatility as follows:

$$S_{ij}^{\text{g}}(H) = \left( \frac{\tilde{\theta}_{ji}^{\text{g}}(H)}{\sum_{i,k=1}^{N} \tilde{\theta}_{ik}^{\text{g}}(H)} - \frac{\tilde{\theta}_{ij}^{\text{g}}(H)}{\sum_{j,k=1}^{N} \tilde{\theta}_{jk}^{\text{g}}(H)} \right) . 100 \qquad (14)$$

$$= \left( \frac{\tilde{\theta}_{ji}^{\text{g}}(H) - \tilde{\theta}_{ij}^{\text{g}}(H)}{N} \right) . 100 \qquad (15)$$

Similarly, the net pairwise volatility spillover between markets *i* and *j* is represented by the difference between the gross volatility shocks communicated from market *i* to market j included those communicated from *j* to *i*.

*4.4. ARCH Model*

The basic autoregressive conditional heteroscedasticity (ARCH) model is constructed from two equations (a mean equation and a variance equation). The mean equation defines the behaviour of the time series data mean, therefore, the mean equation is the linear regression function, which contains constant and other explanatory variables. In the following equation, the mean function only contains an intercept:

$$y_t = \beta + e_t \tag{16}$$

Considering Equation (16), the time series is expected to vary about its mean ($\beta$) randomly. In this case, the error of the regression is distributed normally and heteroskedastic. The variance of the current error period depends on the information that is revealed in the proceeding period Poon (2005). However, the variance equation defines the error variance behaviour where the variance $e_t$ is given the symbol $h_t$ as follows:

$$h_t = a + a_1 e_{t-1}^2 \tag{17}$$

It is clear from Equation (17) that $h_t$ depends on the squared error in the proceeding time period Bollerslev et al. (1994). Additionally, in the same equation, the parameters have to be positive to ensure the variance $h_t$, is positive. In addition, the large multiplier (LM) test can also be used to examine the presence of ARCH effects in the data, (i.e., whether $\alpha > 0$). However, to carry out this test, we estimate the mean equation, then save and square the estimated residuals, $\hat{e}_t^2$. Table 5 shows the regression result of the squared residuals. Then, for the first order ARCH model, we regress $\hat{e}_t^2$ on the lagged residuals $\hat{e}_{t-1}^2$ and the following constant:

$$\hat{e}_t^2 = y_0 + y_1 \hat{e}_{t-1}^2 + v_t \tag{18}$$

where $v_t$ represents the random term. The null and alternative hypothesis are:

$$H_0: y_1 = 0$$

$$H_1: y_1 \neq 0$$

Table 6 shows the result of the large multiplier (LM) test, which confirms the presence of ARCH in the data. Therefore, the forecasted error variance is an in-sample prediction model essentially based on an estimated variance function as follows:

$$\hat{h}_{t+1} = \hat{a}_0 + \left(r_t - \hat{\beta}_0\right)^2 \tag{19}$$

And the forecast error variance $\left(r_t - \hat{\beta}_0\right)^2$, demonstrates the time period of our sample (2005–2016).

## 5. Empirical Results

### 5.1. Descriptive Statistics

Tables 1 and 2 provide descriptive statistics of return and volatility spillovers. The underlying data consist of twenty-three global currencies vis-à-vis the US dollar and the sample size is 2871. Returns are calculated as a daily change in log price of close data (as described in the data section) and return volatilities as signified in Equation (3). Currencies under research are selected based on the most actively traded globally for both developed and developing countries. The augmented Dicky–Fuller (ADF) test results (Tables 1 and 2) for each currency is statistically significant, which means that the currencies under investigation are stationery. For the returns' series (Table 1), fourteen currencies recorded little negative means denoting slight appreciation (during the sample period) against the US dollar, whereas seven currencies recorded small depreciations including the Swiss franc (CHF), the Singaporean dollar (SGD), the Thai baht (THB), the Hong Kong dollar (HKD), the Saudi Arabian riyal (SAR), the United Arab dirham (AED), and the South African rand (ZAR). Kurtosis coefficients are significantly high for developing countries in both returns and volatility spillovers. These are exciting facts that indicate the data distribution is leptokurtic. A distribution that is leptokurtic is said to have a positive statistical value with higher peaks around the mean as compared with a normal distribution, which in most circumstances leads to thick tails on both sides. This means the risks to the developing countries' currencies are coming from outlier events setting the ground for extreme

remarks to arise. In addition, the root means square deviation of volatility spillovers' series (Table 2) shows a significant dispersion for eight developing countries, which include India, Indonesia, Argentina, Malaysia, Thailand, Mexico, South Africa, and Nigeria. For more elaboration on the data, see Tables 1 and 2 below.

**Table 1.** Descriptive statistics, global foreign exchange market returns, 2005–2016.

| Country | United Kingdom | European Union | Australia | Canada | Switzerland |
|---|---|---|---|---|---|
| Mean | 0.000 | 0.000 | 0.000 | 0.000 | 0.000 |
| Standard Error | 0.005 | 0.006 | 0.008 | 0.006 | 0.007 |
| Kurtosis | 3.230 | 2.023 | 11.717 | 2.861 | 80.611 |
| Skewness | 0.408 | −0.048 | 0.830 | −0.036 | −2.676 |
| Minimum | −0.029 | −0.036 | −0.067 | 0.033 | −0.157 |
| Maximum | 0.039 | 0.029 | 0.095 | 0.158 | 0.095 |
| ADF | −51.4786 ** | −53.4031 ** | −55.7591 ** | −54.8177 ** | −53.7565 ** |
| Country | Japan | Iceland | Czech Republic | Hong Kong | Singapore |
| Mean | 0.000 | 0.000 | 0.000 | 0.000 | 0.000 |
| Standard Error | 0.007 | 0.010 | 0.008 | 0.000 | 0.003 |
| Kurtosis | 4.121 | 56.384 | 3.729 | 265.198 | 4.424 |
| Skewness | −0.127 | 0.238 | 0.222 | −9.076 | 0.057 |
| Minimum | −0.044 | −0.134 | −0.050 | −0.032 | −0.022 |
| Maximum | 0.039 | 0.147 | 0.053 | 0.030 | 0.026 |
| ADF | −58.9361 ** | −55.5139 ** | −54.0658 ** | −44.7012 ** | −54.7277 ** |
| Country | South Korea | Russia | Turkey | India | Indonesia |
| Mean | 0.000 | 0.000 | 0.000 | 0.000 | 0.042 |
| Standard Error | 0.007 | 0.009 | 0.008 | 0.004 | 0.851 |
| Kurtosis | 32.781 | 45.221 | 7.001 | 5.945 | 2729.823 |
| Skewness | 0.408 | 0.736 | 0.788 | 1.172 | 51.701 |
| Minimum | −0.103 | −0.141 | −0.053 | −0.035 | −0.098 |
| Maximum | 0.107 | 0.143 | 0.070 | 0.037 | 97.952 |
| ADF | −50.3963 ** | −50.9994 ** | −53.9350 ** | −52.8286 ** | −54.2572 ** |
| Country | Argentine | Malaysia | Thailand | Mexico | Saudi Arabia |
| Mean | 0.000 | 0.000 | 0.000 | 0.000 | 0.000 |
| Standard Error | 0.007 | 0.004 | 0.005 | 0.007 | 0.012 |
| Kurtosis | 1657.464 | 5.182 | 149.717 | 13.351 | 42.832 |
| Skewness | 36.964 | −0.369 | 1.659 | 0.962 | 0.568 |
| Minimum | −0.031 | −0.035 | −0.104 | −0.061 | −0.133 |
| Maximum | 0.355 | 0.029 | 0.115 | 0.081 | 0.153 |
| ADF | −36.8414 ** | −53.5359 ** | −53.5815 ** | −23.8200 ** | −53.5792 ** |
| Country | United A Emirates | South Africa | Nigeria | | |
| Mean | 0.000 | 0.000 | 0.025 | | |
| Standard Error | 0.008 | 0.011 | 1.385 | | |
| Kurtosis | 77.821 | 25.199 | 2870.718 | | |
| Skewness | 0.769 | 1.691 | 53.572 | | |
| Minimum | −0.108 | −0.065 | −0.986 | | |
| Maximum | 0.122 | 0.175 | 74.250 | | |
| ADF | −53.5681 ** | −28.1001 ** | −37.4842 ** | | |

Notes: Returns are in real terms and measured by calculating the daily change in the log price of close data and the sample size is 2871. Asterisk (s) denotes the statistical significance of the Augment Dicky-Fuller (ADF) test as follows: ** $p < 0.05$.

**Table 2.** Descriptive statistics, global foreign exchange market volatility, 2005–2016.

| Country | United Kingdom | European Union | Australia | Canada | Switzerland |
|---|---|---|---|---|---|
| **Mean** | 0.000 | 0.000 | 0.002 | 0.000 | 0.000 |
| **Standard Error** | 0.000 | 0.002 | 0.072 | 0.000 | 0.009 |
| **Kurtosis** | 111.561 | 2866.973 | 1433.442 | 107.130 | 2802.957 |
| **Skewness** | 8.004 | 53.520 | 37.873 | 7.968 | 52.685 |
| **Minimum** | 0.000 | 0.000 | 0.000 | 0.000 | 0.000 |
| **Maximum** | 0.002 | 0.150 | 2.765 | 0.002 | 0.506 |
| **ADF** | −31.2667 ** | −53.5757 ** | −30.9404 ** | −32.0489 ** | −53.5742 ** |
| **Country** | **Japan** | **Iceland** | **Czech Republic** | **Hong Kong** | **Singapore** |
| **Mean** | 0.000 | 0.000 | 0.000 | 0.000 | 0.000 |
| **Standard Error** | 0.000 | 0.001 | 0.000 | 0.000 | 0.000 |
| **Kurtosis** | 259.765 | 1429.986 | 65.781 | 760.508 | 709.547 |
| **Skewness** | 12.947 | 35.395 | 6.512 | 25.702 | 20.668 |
| **Minimum** | 0.000 | 0.000 | 0.000 | 0.000 | 0.000 |
| **Maximum** | 0.003 | 0.088 | 0.003 | 0.000 | 0.001 |
| **ADF** | −42.3771 ** | 25.7536 ** | −30.9438 ** | −15.8937 ** | −28.6243 ** |
| **Country** | **South Korea** | **Russia** | **Turkey** | **India** | **Indonesia** |
| **Mean** | 0.001 | 0.003 | 0.430 | 0.003 | 0.191 |
| **Standard Error** | 0.088 | 0.155 | 23.055 | 0.128 | 2.665 |
| **Kurtosis** | 2871.851 | 2871.755 | 2871.999 | 1214.471 | 226.509 |
| **Skewness** | 53.588 | 53.587 | 53.591 | 34.377 | 14.893 |
| **Minimum** | 0.000 | 0.000 | 0.000 | 0.000 | 0.000 |
| **Maximum** | 4.751 | 8.310 | 1235.575 | 4.7415 | 42.769 |
| **ADF** | −53.5699 ** | −53.5818 ** | −53.5817 ** | −53.6088 ** | −19.8196 ** |
| **Country** | **Argentine** | **Malaysia** | **Thailand** | **Mexico** | **Saudi Arabia** |
| **Mean** | 0.000 | 0.000 | 0.001 | 0.000 | 0.000 |
| **Standard Error** | 0.000 | 0.004 | 0.088 | 0.000 | 0.000 |
| **Kurtosis** | 38.627 | 2843.605 | 2871.925 | 658.920 | 2785.065 |
| **Skewness** | 5.767 | 53.194 | 53.589 | 22.598 | 52.431 |
| **Minimum** | 0.000 | 0.000 | 0.000 | 0.000 | 0.000 |
| **Maximum** | 0.002 | 0.246 | 0.726 | 0.014 | 0.029 |
| **ADF** | −36.8414 ** | −53.5359 ** | −53.5815 ** | −23.8200 ** | −53.5792 ** |
| **Country** | **United A Emirates** | **South Africa** | **Nigeria** | | |
| **Mean** | 0.000 | 0.000 | 0.025 | | |
| **Standard Error** | 0.000 | 0.021 | 0.541 | | |
| **Kurtosis** | 2854.287 | 2868.012 | 750.063 | | |
| **Skewness** | 53.347 | 53.535 | 25.985 | | |
| **Minimum** | 0.000 | 0.000 | 0.000 | | |
| **Maximum** | 0.003 | 1.161 | 18.821 | | |
| **ADF** | −53.5681 ** | −28.1001 ** | −37.4842 ** | | |

Notes: Volatilities are for daily spot closing returns. We employ high-frequency intra-day data (high, low, opening, and closing) to obtain the returns volatilities using Equation (3) described above. The sample size is 2871, consult text for more elaboration. Asterisk (s) denotes the statistical significance of the Augment Dicky-Fuller (ADF) test as follows: ** $p < 0.05$.

### 5.2. Return and Volatility Spillovers: Static Analysis (Spillover Tables)

The spillover index methodology that we applied in this paper is comprised of two steps. Firstly, we provide a full static-sample analysis. Secondly, we successively proceed to interpret the dynamic rolling-sample version. By employing the spillover index, we extract return and volatility spillovers throughout the entire sample (2005–2016). Thus,

we present the spillover indexes for both return and volatility in Tables 3 and 4. The variables $(i, j)$ placed under each table represent the contribution projected to the variance of the 70-day-ahead real foreign exchange (returns Table 1 and volatility Table 2) forecast error of country $i$ coming from innovations to the foreign exchange (returns Table 1 and volatility Table 2) of country $j$.

In both tables, the lower corner of the first column from the right sums the "contributions from others" and similarly from the left sums the "contribution to others." The spillover tables are designed to describe the input and output decomposition of the spillover index. Both products "input and output" help to successfully scrutinise the effect of return and volatility spillovers of global foreign exchange markets across developed and developing countries. With regard to return spillovers (Table 3), touching on developed countries' "contribution to others", we observe that the GBP and the EUR are responsible for the most significant shares of the error variance in forecasting 70-day-ahead, totaling 102 and 100 per cent, respectively. In addition, some developing countries receive significant ''return contribution'' from the developed countries such as Thailand (100%) and Mexico (75%).

Moreover, due to the single European market, return spillovers amongst developed countries are sizeable and positive. This means that there are tremendous cross-market interconnectedness and financial interdependence amid developed countries. Considering the global foreign exchange volatility spillovers (Table 4), it is clear that developed countries contribute significantly to their "own" total volatility spillover. This result is in line with the argument that the currency crisis tends to be regional Glick and Rose (1998) and Yarovaya et al. (2016). The results also show that intra-regional volatility spillovers' transmission tends to be significantly higher than the inter-regional volatility spillovers. Our finding is also in line with Melvin and Melvin (2003), Cai et al. (2008), and Baruník et al. (2017) that significant volatility spillovers are transmitted amid currencies within a particular market.

From Table 4, we find that the pound sterling, Euro, and the Australian dollar are the main contributors of volatility spillovers to others. Again, the result is in line with the findings presented by Antonakakis (2012) and Baruník et al. (2017) who found the GBP and the EUR to be the dominant net transmitters and receivers of volatility spillovers during the period (2000–2013). Following the discussion of the static version of volatility spillovers' transmission across global foreign exchange markets, a key finding is that developed countries contribute substantially to the total volatility transmitted (that is, contributions to others) and received (that is, contributions from others).

So far, we have shown evidence of return and volatility spillovers based on the static version analysis of the spillover indexes presented in Table 3 (return) and Table 4 (volatility). The indexes of 15.1% (for return) and 26.5% (volatility) represent the extracted cross-country spillovers for the full sample (January 2005–July 2016). This means virtually 26.5% of the forecast error variance comes from the spillovers. Aside from scrutinising the broader static effect of return and volatility spillovers across the global foreign exchange markets, we now turn to provide a different fashion of the dynamic movement of return and volatility spillover effect.

**Table 3.** Spillover table. Global foreign exchange (FX) market return, 31 May 2005–1 June 2016.

| | | From | | | | | | | | | | | | | | | | | | | | | | | From Others |
|---|---|---|---|---|---|---|---|---|---|---|---|---|---|---|---|---|---|---|---|---|---|---|---|---|---|
| | | UK | EU | AUS | CAN | CHE | JPN | ISL | CZE | HKG | SGP | KOR | RUS | TUR | IND | IDN | ARG | MYS | THA | MEX | SAU | ARE | ZAF | NGA | |
| | UK | **99.0** | 0.0 | 0.0 | 0.4 | 0.0 | 0.1 | 0.0 | 0.0 | 0.0 | 0.0 | 0.1 | 0.0 | 0.0 | 0.0 | 0.0 | 0.0 | 0.0 | 0.0 | 0.1 | 0.1 | 0.0 | 0.0 | 0.0 | 1 |
| | EU | 0.0 | **99.9** | 0.0 | 0.0 | 0.0 | 0.0 | 0.0 | 0.0 | 0.0 | 0.0 | 0.0 | 0.0 | 0.0 | 0.0 | 0.0 | 0.0 | 0.0 | 0.0 | 0.0 | 0.0 | 0.0 | 0.0 | 0.0 | 0 |
| | AUS | 0.0 | 0.0 | **99.3** | 0.0 | 0.0 | 0.1 | 0.0 | 0.1 | 0.0 | 0.0 | 0.0 | 0.0 | 0.0 | 0.0 | 0.0 | 0.0 | 0.0 | 0.5 | 0.0 | 0.0 | 0.0 | 0.0 | 0.0 | 1 |
| | CAN | 0.7 | 0.0 | 0.0 | **69.1** | 0.0 | 11.6 | 10.3 | 4.9 | 0.4 | 0.5 | 0.5 | 0.0 | 0.0 | 0.0 | 0.0 | 0.4 | 0.0 | 0.3 | 1.2 | 0.0 | 0.2 | 0.0 | 0.0 | 31 |
| | CHE | 0.0 | 0.0 | 0.0 | 0.0 | **100** | 0.0 | 0.0 | 0.0 | 0.0 | 0.0 | 0.0 | 0.0 | 0.0 | 0.0 | 0.0 | 0.0 | 0.0 | 0.0 | 0.0 | 0.0 | 0.0 | 0.0 | 0.0 | 0 |
| | JPN | 0.4 | 0.0 | 0.0 | 11.4 | 0.0 | **75.8** | 0.9 | 6.0 | 0.5 | 0.5 | 0.9 | 0.0 | 0.0 | 0.0 | 0.0 | 0.2 | 0.0 | 0.3 | 3.1 | 0.0 | 0.0 | 0.0 | 0.0 | 24 |
| | ISL | 0.1 | 0.0 | 0.0 | 0.3 | 0.0 | 3.8 | **88.2** | 0.2 | 0.2 | 0.1 | 0.1 | 0.0 | 0.0 | 0.0 | 0.0 | 0.2 | 0.0 | 0.5 | 6.3 | 0.0 | 0.0 | 0.0 | 0.0 | 12 |
| | CZE | 0.3 | 0.0 | 0.0 | 22.1 | 0.1 | 11.3 | 1.1 | **61.5** | 0.3 | 1.7 | 0.1 | 0.0 | 0.0 | 0.0 | 0.0 | 0.3 | 0.0 | 0.5 | 0.5 | 0.0 | 0.0 | 0.0 | 0.0 | 38 |
| | HKG | 0.1 | 0.0 | 0.0 | 0.8 | 0.0 | 0.5 | 0.1 | 0.2 | **97.8** | 0.0 | 0.0 | 0.0 | 0.0 | 0.0 | 0.0 | 0.0 | 0.0 | 0.1 | 0.3 | 0.0 | 0.0 | 0.0 | 0.0 | 2 |
| To | SGP | 0.3 | 0.0 | 0.0 | 5.8 | 0.0 | 3.8 | 0.4 | 7.8 | 0.2 | **80.9** | 0.1 | 0.0 | 0.0 | 0.0 | 0.0 | 0.1 | 0.0 | 0.1 | 0.4 | 0.0 | 0.0 | 0.0 | 0.0 | 19 |
| | KOR | 0.0 | 0.0 | 0.0 | 0.1 | 0.0 | 0.4 | 0.2 | 0.1 | 22.7 | 0.0 | **76.0** | 0.0 | 0.0 | 0.0 | 0.0 | 0.0 | 0.0 | 0.3 | 0.1 | 0.0 | 0.0 | 0.0 | 0.0 | 24 |
| | RUS | 0.0 | 0.0 | 0.0 | 0.0 | 0.0 | 0.0 | 0.0 | 0.0 | 0.0 | 0.0 | 0.0 | **99.6** | 0.0 | 0.0 | 0.0 | 0.3 | 0.0 | 0.0 | 0.0 | 0.0 | 0.0 | 0.0 | 0.0 | 0 |
| | TUR | 0.0 | 0.0 | 0.0 | 0.0 | 0.0 | 0.0 | 0.0 | 0.1 | 0.1 | 0.0 | 0.0 | 0.0 | **99.7** | 0.0 | 0.0 | 0.0 | 0.0 | 0.0 | 0.0 | 0.0 | 0.0 | 0.0 | 0.0 | 0 |
| | IND | 0.0 | 0.0 | 0.0 | 0.0 | 0.0 | 0.0 | 0.0 | 0.0 | 0.0 | 0.0 | 0.0 | 0.0 | 0.0 | **99.9** | 0.0 | 0.0 | 0.0 | 0.0 | 0.0 | 0.0 | 0.0 | 0.0 | 0.0 | 0 |
| | IDN | 0.0 | 0.0 | 0.0 | 0.0 | 0.0 | 0.0 | 0.0 | 0.0 | 0.0 | 0.0 | 0.1 | 0.0 | 0.0 | 0.0 | **99.7** | 0.0 | 0.0 | 0.1 | 0.0 | 0.0 | 0.0 | 0.0 | 0.0 | 0 |
| | ARG | 0.2 | 0.0 | 0.0 | 1.6 | 0.0 | 5.4 | 0.5 | 2.7 | 0.3 | 0.2 | 0.1 | 0.0 | 0.0 | 0.0 | 0.0 | **85.1** | 0.0 | 1.4 | 0.8 | 0.0 | 0.0 | 1.6 | 0.0 | 15 |
| | MYS | 0.0 | 0.0 | 0.0 | 0.0 | 0.0 | 0.0 | 0.0 | 0.0 | 0.0 | 0.0 | 0.0 | 0.0 | 0.0 | 0.0 | 0.0 | 0.0 | **99.9** | 0.0 | 0.0 | 0.0 | 0.0 | 0.0 | 0.0 | 0 |
| | THA | 0.0 | 0.0 | 0.0 | 0.1 | 0.0 | 0.4 | 0.2 | 0.1 | 22.7 | 0.0 | 76.0 | 0.0 | 0.0 | 0.0 | 0.0 | 0.0 | 0.0 | **0.3** | 0.1 | 0.0 | 0.0 | 0.0 | 0.0 | 100 |
| | MEX | 0.1 | 0.0 | 0.0 | 0.5 | 0.0 | 5.9 | 63.8 | 1.8 | 0.1 | 0.3 | 0.2 | 0.0 | 0.0 | 0.0 | 0.0 | 0.2 | 0.0 | 0.3 | **24.7** | 0.0 | 0.0 | 0.0 | 0.0 | 75 |
| | SAU | 0.6 | 0.0 | 0.0 | 0.0 | 0.0 | 0.0 | 0.0 | 0.0 | 0.0 | 0.0 | 0.0 | 0.0 | 0.0 | 0.0 | 0.0 | 0.0 | 0.0 | 0.0 | 0.0 | **99.3** | 0.0 | 0.0 | 0.0 | 2 |
| | ARE | 0.0 | 0.0 | 0.0 | 0.0 | 0.0 | 0.0 | 0.0 | 0.0 | 0.0 | 0.0 | 0.0 | 0.0 | 0.0 | 0.0 | 0.0 | 0.1 | 0.0 | 0.0 | 0.0 | 0.0 | **99.8** | 0.0 | 0.0 | 0 |
| | ZAF | 0.0 | 0.0 | 0.0 | 0.0 | 0.0 | 0.0 | 0.0 | 0.0 | 0.0 | 0.0 | 0.0 | 0.0 | 0.0 | 0.0 | 0.0 | 1.4 | 0.0 | 0.5 | 0.0 | 0.0 | 0.0 | **98.0** | 0.0 | 2 |
| | NGA | 0.0 | 0.0 | 0.0 | 0.1 | 0.0 | 0.1 | 0.1 | 0.0 | 0.0 | 0.0 | 0.0 | 0.0 | 0.0 | 0.0 | 0.0 | 0.0 | 0.0 | 0.1 | 0.2 | 0.0 | 0.0 | 0.0 | **99.3** | 1 |
| Contribution to Others | | 3 | 0 | 0 | 45 | 0 | 43 | 78 | 24 | 47 | 3 | 78 | 0 | 0 | 0 | 0 | 3 | 0 | 5 | 13 | 0 | 0 | 2 | 0 | 347 |
| Contribution Including Own | | 102 | 100 | 99 | 114 | 100 | 119 | 166 | 86 | 145 | 84 | 154 | 100 | 100 | 100 | 100 | 89 | 100 | 5 | 38 | 99 | 100 | 100 | 99 | 15.1% |

Note: The fundamental variance decomposition is based on weekly (VAR) of order 2 identified using Cholesky factorisation. The value of $(i, j)$ variables is the estimated contribution to the variance of the 70-day-ahead real foreign exchange (FX) return forecast error of country $i$ coming innovations to real FX returns of country $j$.

**Table 4.** Spillover table. Global foreign exchange (FX) market volatility, 31 May 2005–1 June 2016.

| | | | | | | | | | | | | From | | | | | | | | | | | | | |
|---|---|---|---|---|---|---|---|---|---|---|---|---|---|---|---|---|---|---|---|---|---|---|---|---|---|
| | | UK | EU | AUS | CAN | JPN | CHE | ISL | HKG | CZE | SGP | KOR | RUS | TUR | IND | IDN | ARG | MYS | THA | MEX | SAU | ARE | ZAF | NGA | From Others |
| To | UK | **97.4** | 0.0 | 0.2 | 0.4 | 0.0 | 0.1 | 0.2 | 0.0 | 0.6 | 0.1 | 0.0 | 0.2 | 0.3 | 0.0 | 0.0 | 0.0 | 0.1 | 0.0 | 0.2 | 0.0 | 0.1 | 0.0 | 0.1 | **3** |
| | EU | 39.4 | **59.0** | 0.3 | 0.0 | 0.0 | 0.2 | 0.1 | 0.1 | 0.2 | 0.0 | 0.1 | 0.0 | 0.2 | 0.0 | 0.0 | 0.0 | 0.0 | 0.0 | 0.1 | 0.1 | 0.0 | 0.0 | 0.0 | **41** |
| | AUS | 24.8 | 6.2 | **62.5** | 1.5 | 0.0 | 0.3 | 0.7 | 0.0 | 0.2 | 0.2 | 0.1 | 0.1 | 0.4 | 0.1 | 0.0 | 0.0 | 0.1 | 0.0 | 0.4 | 0.1 | 0.1 | 0.2 | 0.0 | **37** |
| | CAN | 24.6 | 5.4 | 15.0 | **53.2** | 0.0 | 0.1 | 0.1 | 0.0 | 0.4 | 0.0 | 0.0 | 0.1 | 0.3 | 0.1 | 0.0 | 0.0 | 0.2 | 0.0 | 0.1 | 0.1 | 0.0 | 0.0 | 0.0 | **47** |
| | JPN | 0.1 | 0.1 | 0.1 | 0.1 | **98.1** | 0.0 | 0.4 | 0.0 | 0.1 | 0.2 | 0.1 | 0.1 | 0.1 | 0.0 | 0.1 | 0.0 | 0.0 | 0.0 | 0.1 | 0.0 | 0.2 | 0.0 | 0.0 | **2** |
| | CHE | 17.8 | 26.8 | 0.4 | 0.6 | 0.0 | **53.0** | 0.0 | 0.3 | 0.3 | 0.1 | 0.1 | 0.0 | 0.1 | 0.1 | 0.0 | 0.0 | 0.2 | 0.0 | 0.0 | 0.1 | 0.0 | 0.0 | 0.0 | **47** |
| | ISL | 14.4 | 10.7 | 1.2 | 0.3 | 0.1 | 0.4 | **69.4** | 0.4 | 0.1 | 0.4 | 0.3 | 0.0 | 0.1 | 0.0 | 0.0 | 0.1 | 0.1 | 0.1 | 1.9 | 0.0 | 0.0 | 0.1 | 0.0 | **31** |
| | HKG | 0.9 | 1.0 | 1.5 | 0.1 | 0.1 | 0.0 | 0.3 | **94.5** | 0.1 | 0.0 | 0.2 | 0.1 | 0.1 | 0.0 | 0.0 | 0.0 | 0.2 | 0.2 | 0.1 | 0.0 | 0.0 | 0.1 | 0.3 | **5** |
| | CZE | 33.7 | 38.8 | 0.8 | 0.4 | 0.0 | 0.1 | 0.1 | 0.0 | **25.3** | 0.0 | 0.1 | 0.1 | 0.1 | 0.0 | 0.0 | 0.0 | 0.0 | 0.0 | 0.2 | 0.1 | 0.0 | 0.1 | 0.0 | **75** |
| | SGP | 26.9 | 14.2 | 10.2 | 1.3 | 0.1 | 0.4 | 0.2 | 1.9 | 0.5 | **43.1** | 0.1 | 0.1 | 0.1 | 0.1 | 0.1 | 0.0 | 0.2 | 0.0 | 0.3 | 0.3 | 0.0 | 0.1 | 0.0 | **57** |
| | KOR | 8.1 | 1.7 | 9.2 | 1.3 | 0.1 | 0.1 | 0.1 | 0.2 | 0.5 | 7.1 | **64.7** | 0.0 | 2.2 | 0.4 | 0.1 | 0.1 | 0.4 | 0.0 | 1.7 | 0.1 | 0.0 | 1.0 | 0.1 | **35** |
| | RUS | 0.1 | 0.2 | 0.1 | 0.1 | 0.1 | 0.1 | 0.1 | 0.0 | 0.1 | 0.1 | 0.1 | **98.1** | 0.0 | 0.1 | 0.1 | 0.0 | 0.1 | 0.0 | 0.4 | 0.0 | 0.0 | 0.0 | 0.0 | **2** |
| | TUR | 13.2 | 4.3 | 10.2 | 3.5 | 0.1 | 1.2 | 0.6 | 0.0 | 1.8 | 1.1 | 0.4 | 0.1 | **61.9** | 0.0 | 0.0 | 0.0 | 0.0 | 0.0 | 0.9 | 0.1 | 0.0 | 0.5 | 0.0 | **38** |
| | IND | 6.8 | 1.6 | 4.8 | 0.9 | 0.3 | 0.1 | 0.3 | 0.2 | 0.1 | 2.9 | 1.7 | 0.2 | 2.0 | **76.1** | 0.2 | 0.0 | 0.3 | 0.0 | 1.1 | 0.1 | 0.0 | 0.3 | 0.0 | **24** |
| | IDN | 0.0 | 0.1 | 0.0 | 0.1 | 0.3 | 0.1 | 0.0 | 0.3 | 0.0 | 0.1 | 0.0 | 0.2 | 0.1 | 0.0 | **98.2** | 0.0 | 0.0 | 0.0 | 0.3 | 0.0 | 0.0 | 0.0 | 0.0 | **2** |
| | ARG | 0.1 | 0.0 | 0.4 | 0.1 | 0.1 | 0.0 | 0.0 | 0.0 | 0.0 | 0.2 | 0.1 | 0.1 | 0.0 | 0.0 | 0.2 | **98.3** | 0.1 | 0.0 | 0.1 | 0.0 | 0.0 | 0.0 | 0.0 | **2** |
| | MYS | 7.2 | 3.8 | 5.3 | 2.1 | 0.1 | 0.2 | 0.1 | 0.8 | 0.2 | 13.0 | 2.1 | 0.1 | 1.2 | 2.5 | 0.2 | 0.1 | **59.6** | 0.0 | 1.2 | 0.1 | 0.0 | 0.2 | 0.0 | **40** |
| | THA | 1.8 | 1.3 | 1.0 | 0.1 | 0.1 | 0.2 | 0.1 | 0.3 | 0.1 | 3.3 | 0.2 | 0.0 | 0.4 | 0.8 | 0.1 | 0.0 | 0.6 | **89.4** | 0.0 | 0.0 | 0.0 | 0.0 | 0.0 | **11** |
| | MEX | 14.4 | 2.7 | 8.7 | 8.4 | 0.0 | 1.1 | 0.2 | 0.2 | 2.0 | 3.6 | 0.4 | 0.1 | 4.7 | 0.3 | 0.3 | 0.0 | 0.2 | 0.0 | **52.7** | 0.1 | 0.0 | 0.0 | 0.0 | **47** |
| | SAU | 0.1 | 0.1 | 0.0 | 0.0 | 0.0 | 0.0 | 0.1 | 0.0 | 0.0 | 0.0 | 0.1 | 0.0 | 0.2 | 0.1 | 0.0 | 0.1 | 0.1 | 0.0 | 0.0 | **98.5** | 0.6 | 0.0 | 0.0 | **2** |
| | ARE | 0.0 | 0.0 | 0.1 | 0.1 | 0.3 | 0.0 | 0.0 | 0.0 | 0.0 | 0.1 | 0.0 | 0.0 | 0.1 | 0.1 | 0.0 | 0.0 | 0.0 | 0.0 | 0.0 | 0.6 | **98.6** | 0.0 | 0.0 | **1** |
| | ZAF | 18.5 | 5.1 | 10.7 | 4.0 | 0.1 | 0.3 | 0.7 | 0.1 | 2.1 | 2.6 | 0.2 | 0.1 | 9.4 | 0.0 | 0.0 | 0.0 | 0.0 | 0.0 | 5.7 | 0.0 | 0.0 | **39.5** | 0.0 | **60** |
| | NGA | 0.1 | 0.2 | 0.0 | 0.0 | 0.1 | 0.2 | 0.0 | 0.0 | 0.0 | 0.2 | 0.0 | 0.0 | 0.0 | 0.0 | 0.0 | 0.0 | 0.3 | 0.0 | 0.1 | 0.0 | 0.0 | 0.0 | **98.8** | **1** |
| Contribution to Others | | 253 | 124 | 80 | 26 | 2 | 5 | 5 | 5 | 9 | 35 | 6 | 2 | 23 | 5 | 2 | 1 | 3 | 1 | 16 | 2 | 1 | 3 | 1 | **610** |
| Contribution Including Own | | 351 | 183 | 143 | 79 | 100 | 58 | 75 | 99 | 35 | 78 | 71 | 100 | 85 | 81 | 100 | 99 | 63 | 90 | 68 | 100 | 100 | 42 | 99 | **26.5%** |

Note: The fundamental variance decomposition is based on daily (VAR) of order 2 identified using Cholesky factorisation. The value of $(i, j)$ variables is the estimated contribution to the variance of the 70-day-ahead foreign exchange volatility forecast error of country $i$ coming from innovation to the foreign exchange volatility of country $j$.

*5.3. Return and Volatility Spillovers: Dynamic Analysis (Spillover Plots)*

To address the extent of the spillover effect between developed and developing countries we use 200-day rolling samples, which is about six months. The 200-day rolling sample was used to demonstrate the spillover variations over time between developed and developing countries since the data we used spans over the time period 2005–2016. The dynamic movement of return and volatility spillovers is designed to capture the effect of the potential recurring movement of spillovers by using returns and volatility indexes shown in Tables 3 and 4. The indexes are the sums of all variance decompositions represented in the form of "contribution to others." Employing the indexes, we estimate the model to scrutinise the evolution of global foreign exchange markets during the time period of the sample (2005–2016). Hence, we capture the magnitude and disparities of the spillovers for return and volatility, which we present graphically in the form of spillover plots.

The era of the 2000s, which began with a recession, mainly in developed countries across the European Union and the USA undisputedly, documented painful economic events in our history, in particular, the 2008 global financial turmoil. Thus, Figure 1 (return spillovers) captures some of the critical events, whereas Figure 2 (volatility spillovers) appears to be the most eventful. Interestingly, the 200-day rolling samples epitomised in Figures 1 and 2 highlight some of the significant economic events that occurred during the time period of the sample (2005–2016). As the estimation window moves towards 2016, we captured the following critical economic events:

(1)  The U.S. housing bubble worries, according to Liebowitz (2009) foreclosure rates, increased by 43 per cent during the second and the fourth quarter of 2006.
(2)  The increase in foreclosures and mortgage default rates reached about 55 per cent (prime) and 80 per cent (subprime), which hugely devalued mortgage-back-securities at the end of 2007, causing a severe credit crunch.
(3)  In addition, in 2007, the British bank Northern Rock collapsed.
(4)  Then, Lehman Brothers, the biggest U.S. investment bank, filed for bankruptcy on 15 September 2008.
(5)  The above events, among others, were followed by the worst financial turmoil (2007–2009) since the great depression (1929–1939), and the Greece debt crisis (December 2009).
(6)  The European sovereign debt crisis occurred in 2009.
(7)  The crude oil prices fell in 2014.
(8)  The leading causes of the Russia financial crisis (2014–2017), according to the Centre for Eastern Studies (OSW) were the tensions between Russia and the west, which led to a sanction war and a dramatic fall in oil prices.
(9)  First signs of Brexit worries began on 23 June 2016, whereby the British pound sterling plunged to its lowest level since 1985.

The graphical illustrations in Figures 1 and 2 highlight important economic events during the time period of the sample (2005–2016). The analysis orchestrated here, visually signals the effect of volatility spillovers across intra-foreign exchange markets. The magnitude and extent of the spillover effect of both returns (Figure 1) and volatility (Figure 2) were significantly marked by the crisis episodes of (2007–09), in particular, the series of European sovereign debt crisis (2009–2014) and the China stock market crash (2015), among others. This means, interestingly, besides volatility spillovers, the contribution of return spillovers is unexpectedly significant enough to show some commonality with volatility spillovers in terms of responding to economic events.

Furthermore, we also observe bursts in total return and volatility spillovers which materialise twice in Figure 1 and four times in Figure 2. The total return's spillover began to decrease slightly after its strong response to the (2007–09) financial turmoil as well as the European sovereign debt crisis in 2009 until the China stock market crash in (2015), whereby it shows a dramatic increase.

On the contrary, volatility spillovers fluctuate with explicit outbursts virtually with every single economic event highlighted during the time period of the full sample (2005–2016). In other words, the volatility spillover plot (Figure 2) depicts the phenomenon of the global systemically important financial institutions from a series of historical defaults involved that are too big to fail nature. To check the robustness of the results regarding rolling window width, forecast horizon, and VAR ordering, we perform spillover plots (Figure 3) using 84-day rolling window width, we also used two different variance decomposition forecast horizons, i.e., a 70-day forecast horizon in Figure 3a and 14-day forecast horizon in Figure 3b. The results are robust even when employing maximum and minimum volatility spillovers across a diversity of alternative VAR ordering using 200-day rolling windows (see Figures 3 and 4).

*5.4. Robustness Analysis*

According to the extent of the above results, the maximum and minimum spillovers (Figure 4) show the variability of the volatility spillovers' magnitude in global foreign exchange markets, which appears to be relatively higher than the return spillovers. Notwithstanding, we find the behaviour of return spillovers in the global currency markets (Figure 1) substantially responding to major economic events. Since we find "contribution to others" mainly dominated by developed countries, in particular, the British pound sterling (GBP), the Euro (EUR), and the Australian dollar (AUD), this means developing countries act as net receivers to return and volatility spillovers.

Furthermore, according to the Bank for International Settlements' (BIS) report (2013), the USD, EUR, GBP, AUD, CAD, JPY, and CHF are the most traded currencies globally, and account for almost 90 per cent of the global foreign exchange turnover. This means that a substantial amount of return and volatility spillovers transmitted across the world during the time period of the full sample (2005–2016) are coming from developed countries. The findings are robust even when employing maximum and minimum volatility spillovers across a diversity of alternative VAR ordering using 200-day rolling windows.

Interestingly, the results highlight the significance of the global foreign exchange markets' spillover channels during crisis periods in several dimensions. First, the results highlight the cyclical bursts in spillovers that occur as a consequence of significant economic events. These include the credit crunch of July 2007, Lehman Brothers collapsed in September 2008, the financial turmoil which created havoc during 2007–2009, the European sovereign debt crisis (2009–2014), and the fall in crude oil prices in 2013.

Secondly, the results highlight the potential magnitude of the spillover effect, in particular, from the default of systemically important financial institutions across the global financial system, which spread jitters from the outset of the U.S. subprime mortgage crisis. Third, the results highlight the size of the shocks which led to bursts in spillovers (see Figures 1–4) suggest strong cross-market interconnectedness. This reflects the definition of contagion presented by Forbes and Rigobon (2002) as a significant increase in cross-market linkages after a shock to one country or group of countries. Fourth, the results also provide significant insights, particularly to the financial regulators, from the perspectives of understanding the effect of spillovers from the default of systemically important financial institutions. Finally, the results also introduce, for investors, the issue of cross-market linkages and economic interdependence during crises periods, whereby volatility spillovers increase substantially.

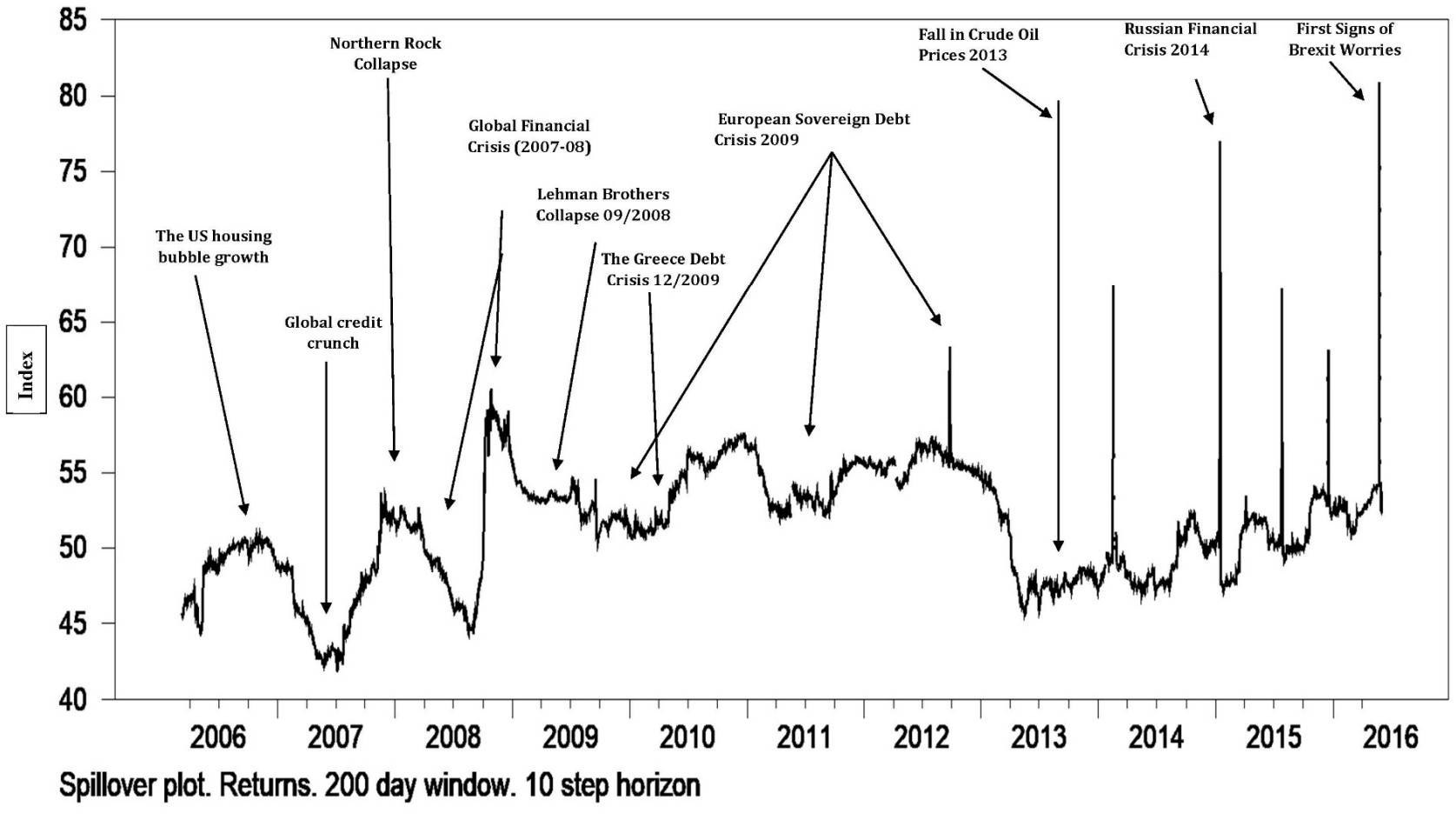

**Ending Date of Window**

**Figure 1.** Spillover plot. Global foreign exchange market returns (2005–2016).

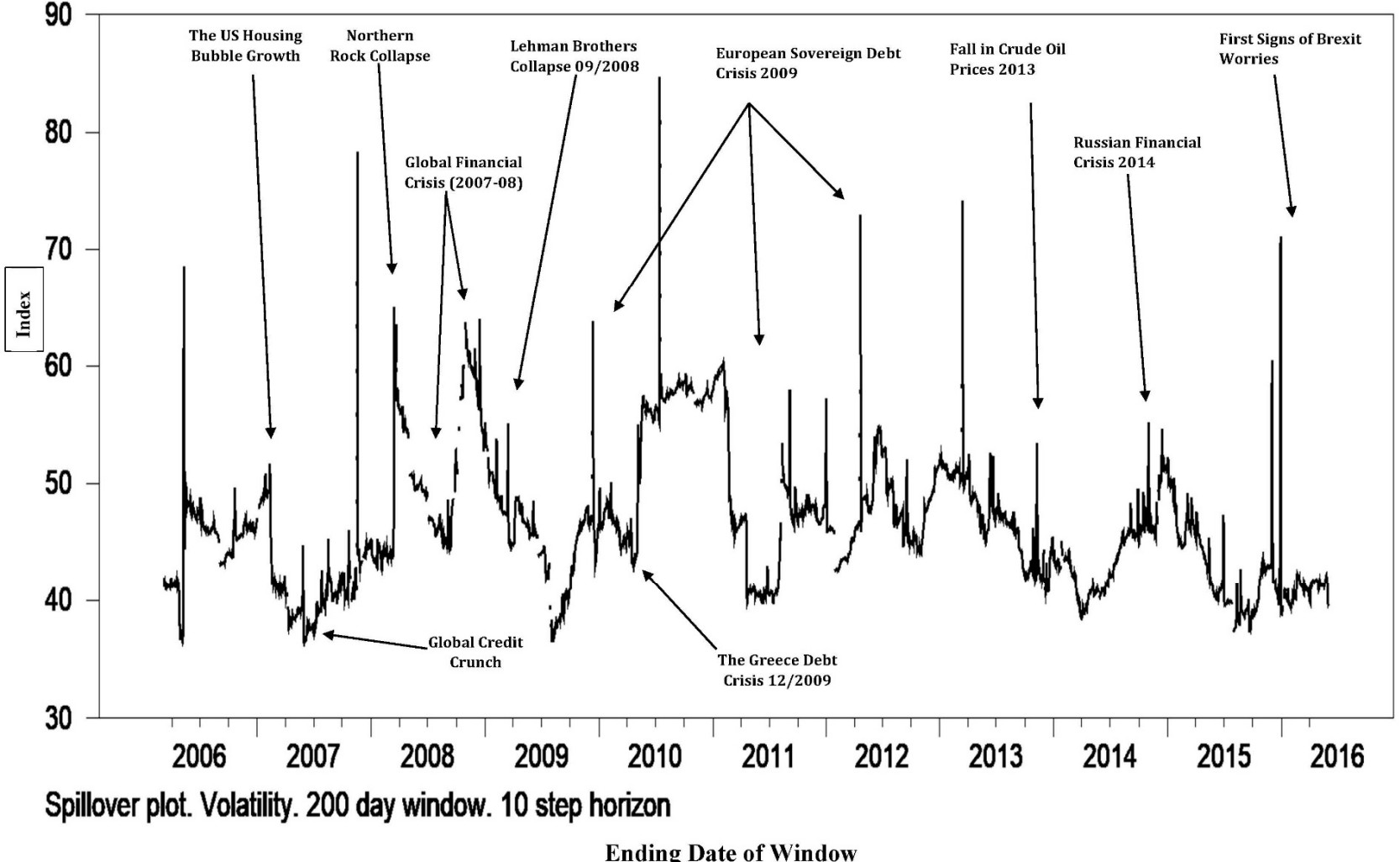

**Figure 2.** Spillover plot. Global foreign exchange market volatility (2005–2016).

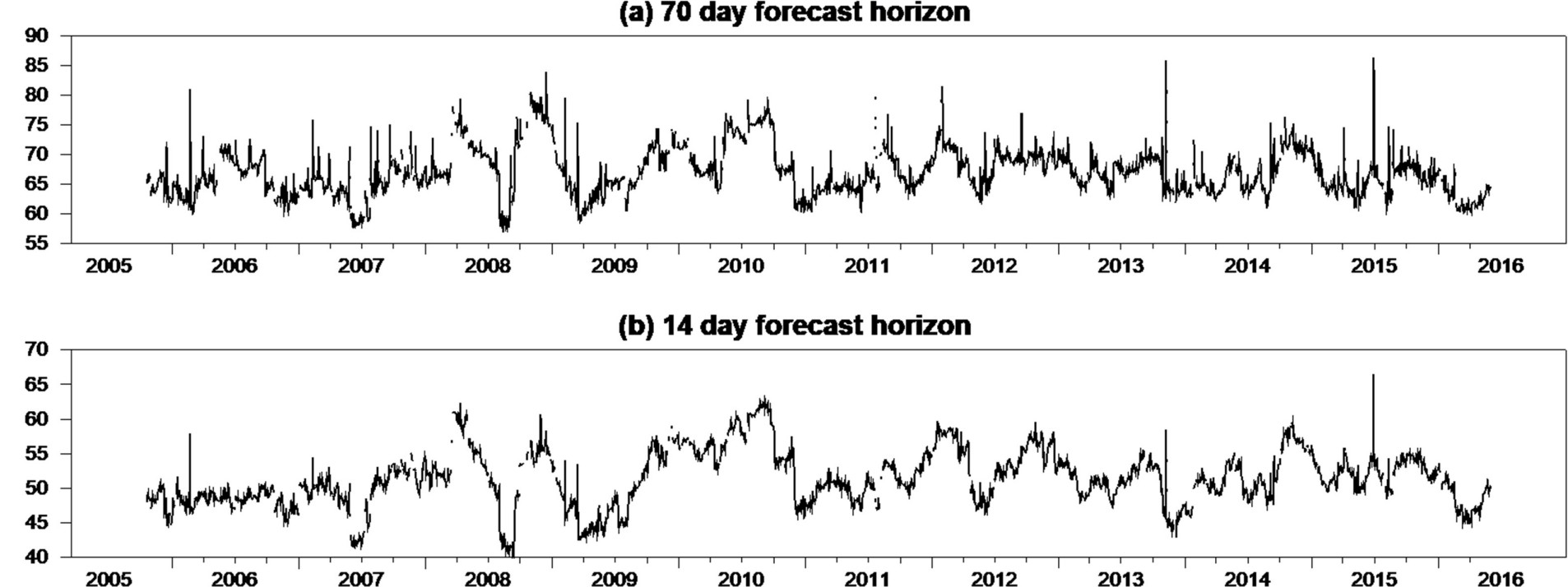

**Figure 3.** Spillover plots. Global foreign exchange markets (2005–2016), 70- and 14-day forecast horizons.

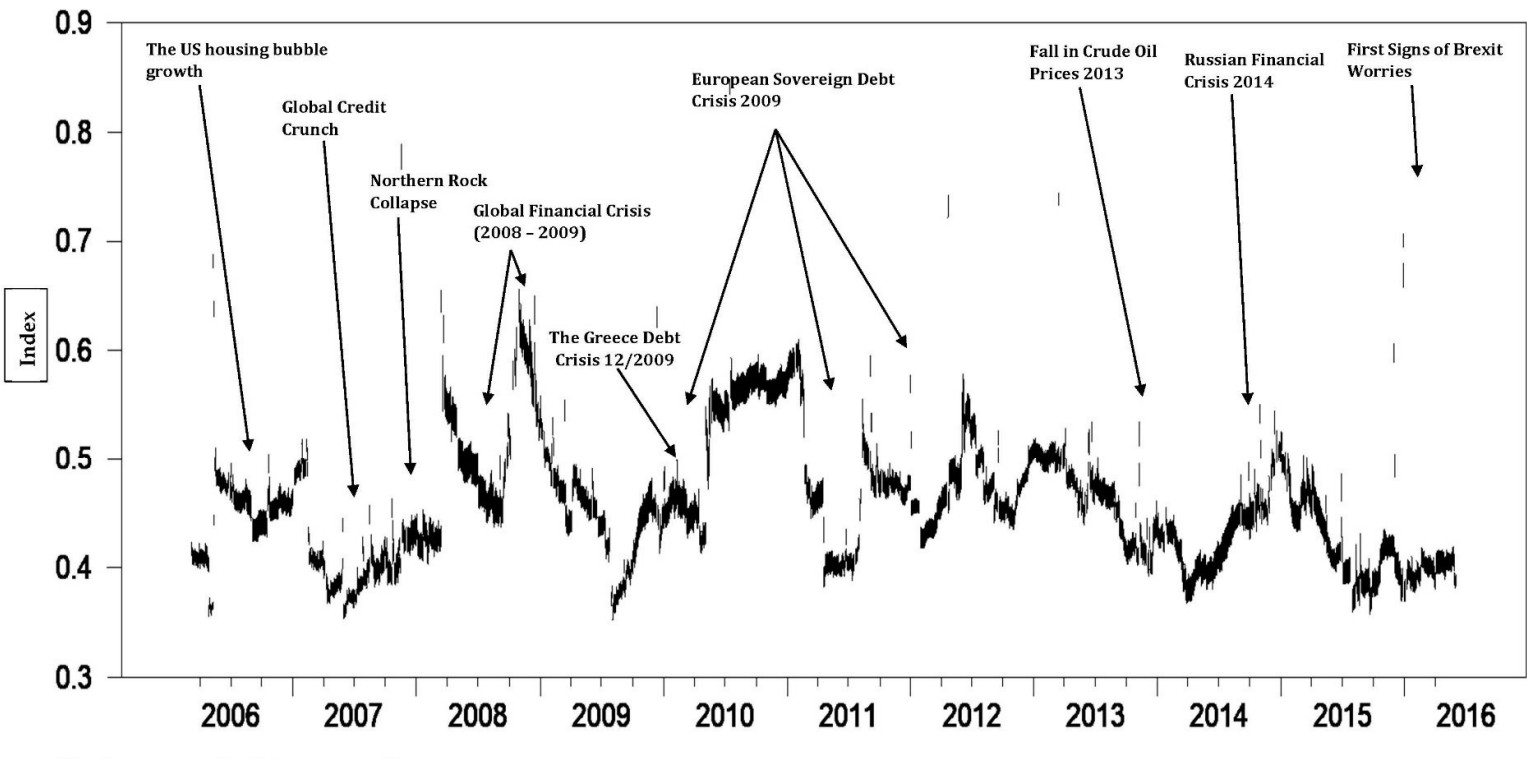

**Figure 4.** Maximum and minimum spillovers. Global foreign exchange markets (2005–2016).

## 6. Time-Varying Volatility Spillovers

In this section, we present the results of the time-varying volatility spillovers among developed and developing countries using autoregressive conditional heteroskedasticity (ARCH). Time-varying volatility helps to investigate sources of significant shifts in the volatility during the time period of our sample (2005–2016). This is because an ARCH model is designed to capture persistence in time-varying volatility based on squared returns Poon (2005). The ARCH model has a unique structure, where "autoregressive" means high volatility tends to persist, "conditional" refers to time-varying or specific point on time, and "heteroskedasticity" refers to non-constant volatility Poon (2005). Before applying the ARCH (1) model, we first generate the squared residuals using regression, which contains only an intercept. Table 5 shows the regression result of the squared residuals. This is because the squared residuals ensure that the conditional variance is positive and, consequently, the leverage effects cannot be captured by the ARCH model Engle (2001b).

**Table 5.** Regression results (squared residuals).

| Variable | Adjusted $t^*$ | | *p*-Value |
|---|---|---|---|
| Ehat2 | 8.12 | | 0.000 |
| No-of-Obs: 2.871 | R-Squared: 0.022 | Adj R-Squared: 0.002 | MSE: $1.3 \times 10^{-7}$ |

Second, we test the data for the presence of ARCH effects using the Box–Pierce large multiplier (LM), which provides the most appropriate results Alexander (2002). Table 6 displays the result of the large multiplier's (LM) test for the presence of ARCH effects in the data.

**Table 6.** LM test results for autoregressive conditional heteroskedasticity (ARCH).

| lags(p) | chi2 | df | Prob > chi2 |
|---|---|---|---|
| 1 | 1 | 64.443 | 0.0000 |

H0, no ARCH effects vs. H1, ARCH (p) disturbance.

The LM results show the null and alternative hypotheses, the statistic and its distribution, and the *p*-value, which indicates the presence of ARCH (p) model disturbance in the data. Thus, we estimate the ARCH (1) model and generate the forecast error variance, which is essentially an in-sample prediction model based on the estimated variance function (see Equation (19) for more details). Table 7 shows the result of the conditional variance of the estimated ARCH (1) model. The conditional variance in the ARCH model is allowed to change over time as a function of past error leaving the unconditional variance constant Bollerslev (1986). Then, we proceed by plotting the forecast error variance against the time period of our sample (2005–2016). Figure 5 shows the result of the ARCH (1) model, which implies that the volatility spillovers from developed countries to the developing countries seem to be specifically strong in 2008.

Thus, the result indicates that the foreign exchange markets' channels between developed and developing countries exhibit time-varying persistence in its conditional volatilities over the crisis period. This result is consistent with the spillover index findings of both the static analysis (Table 4) and the dynamic analysis (Figures 2 and 4). It also shows that all the currencies in the sample from both (developed and developing) countries are characterised by clustering volatility. Our findings indicate that the global foreign exchange market experiences somewhat relatively sedate volatility spillovers from 2005 to 2007. Then, the foreign exchange market's volatility spillovers become much more volatile in 2008–2009. These results are consistent with the dynamic analysis of the spillover indices (Figure 2), which captured the 2008/2009 financial crisis.

**Table 7.** ARCH (1) conditional variance for global foreign exchange market (2005–2016).

| 496 | $2.8 \times 10^{-9}$ | $2.8 \times 10^{-9}$ |
|---|---|---|
| 497 | $2.24 \times 10^{-9}$ | $2.24 \times 10^{-9}$ |
| 498 | $2.99 \times 10^{-9}$ | $2.99 \times 10^{-9}$ |
| 499 | $2.56 \times 10^{-9}$ | $2.56 \times 10^{-9}$ |
| 500 | $4.02 \times 10^{-9}$ | $4.02 \times 10^{-9}$ |

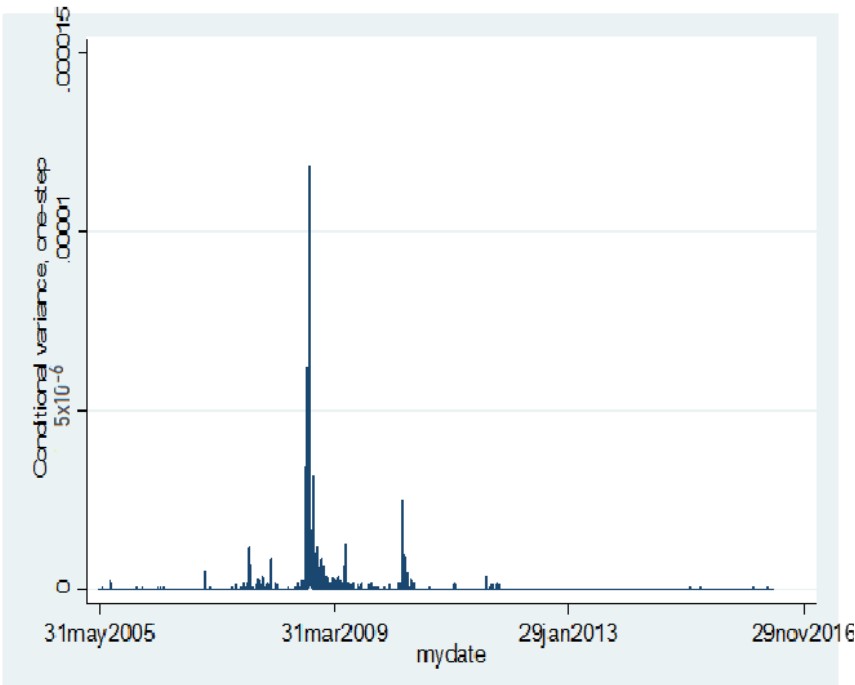

**Figure 5.** Results of the global foreign exchange market time-varying volatility, over the time period 2005–2016, using the ARCH (1) model. Note: This figure shows the persistence in time-varying volatility between developed and developing countries over the time period 2005–2016. The underlying currencies across developed and developing countries, including the most traded globally, are described in the data section.

## 7. Net Spillovers and Net Pairwise Volatility Spillovers

In this section, we present the results of the net spillovers and the net pairwise spillovers between developed and developing countries over the time period of our sample (2005–2016). The key features of the net volatility spillovers show the difference between the gross volatility shocks that are transmitted to and those received from all other markets (Diebold and Yilmaz 2012). Therefore, the net pairwise volatility spillover (Equation (14)) between country i and j is the difference between the gross volatility shocks transmitted from a country i to country j including the transmission from j to i (Diebold and Yilmaz 2012). As shown in Equation (12), the net volatility spillovers offer important information about the amount of volatility in net terms that each country contributes to other countries. Therefore, the main focus point of this section is to calculate the net volatility and the net pairwise volatility spillovers between developed and developing countries. Due to a large number of countries (23) in our sample, only 16 currencies were selected, which we present in Figures 6–9.

During the time period of our sample (2005–2016), there were two major events of net volatility spillovers through the global foreign exchange market, in particular, during the 2008/2009 financial crisis and the European sovereign debt crisis in 2009/2013. However, before the recent financial crisis and the European sovereign debt crisis, the net vol-

atility spillovers between developed and developing countries was relatively low; however, things changed drastically after 2007, when the net volatility spillover from the EUR to the Malaysian ringgit (Figure 8) jumped to 20% in the third quarter of 2008 and 40% in the third quarter of 2009. These results are consistent with the time-varying volatility results, which imply that the foreign exchange market experienced low volatility from 2005 to 2007. The pound sterling (GBP) and the Euro (EUR) (Figures 6–9) both act as giving and receiving of the net volatility transmissions, with almost similar magnitudes across the global foreign exchange markets. This finding supports the static analysis of the spillover index (Table 4) that the pound sterling (GBP) and the Euro (EUR) are the main contributors of volatility spillovers.

The Indonesian rupiah (IDR) also receives a significant amount of volatility spillovers from the Euro (EUR) (Figure 7), especially during the recent financial crisis and the European sovereign debt crisis in 2009/1013. In addition, the Euro (EUR) receives a large amount of volatility spillovers from the Malaysian ringgit (Figure 9), which indicates that developed countries act as receivers and transmitters of volatility spillovers. The Argentine peso (ARS) contributes as well as receives a significant amount of volatility from the Malaysian ringgit (MYR), Figure 9. The net volatility spillover from the pound sterling (GBP) to the Euro (EUR) (Figure 9) seems relatively low while receiving a significant amount of volatility spillover from the Euro (EUR). The fact that the pound sterling (GBP) contributes as well as receives a large amount of volatility spillover from the Euro (EUR) shows the increased link between developed countries in the global foreign exchange market.

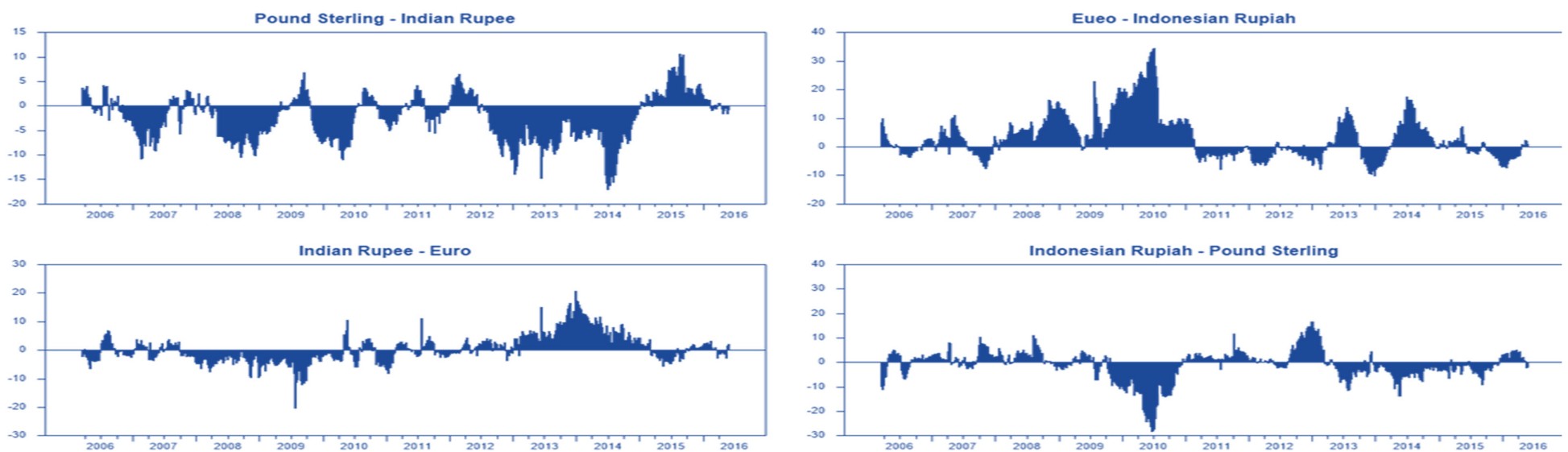

**Figure 6.** This figure shows the net volatility spillovers between developed and developing countries. In particular, the net volatility spillovers from/to the Pound Sterling (GBP), Indian Rupee (INR), Euro (EUR), and the Indonesian Rupiah (IDR).

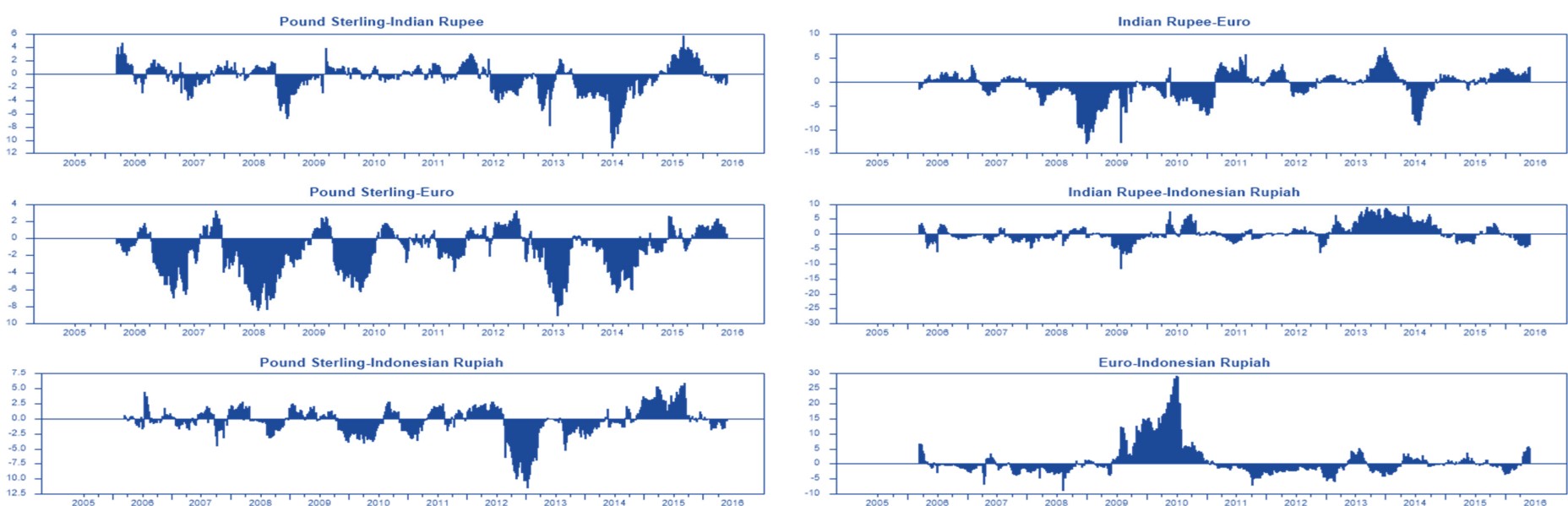

**Figure 7.** This figure shows the net pairwise volatility spillovers between developed and developing countries. In particular, the net pairwise volatility spillovers from/to the Pound Sterling (GBP), Indian Rupee (INR), Euro (EUR), and the Indonesian Rupiah (IDR).

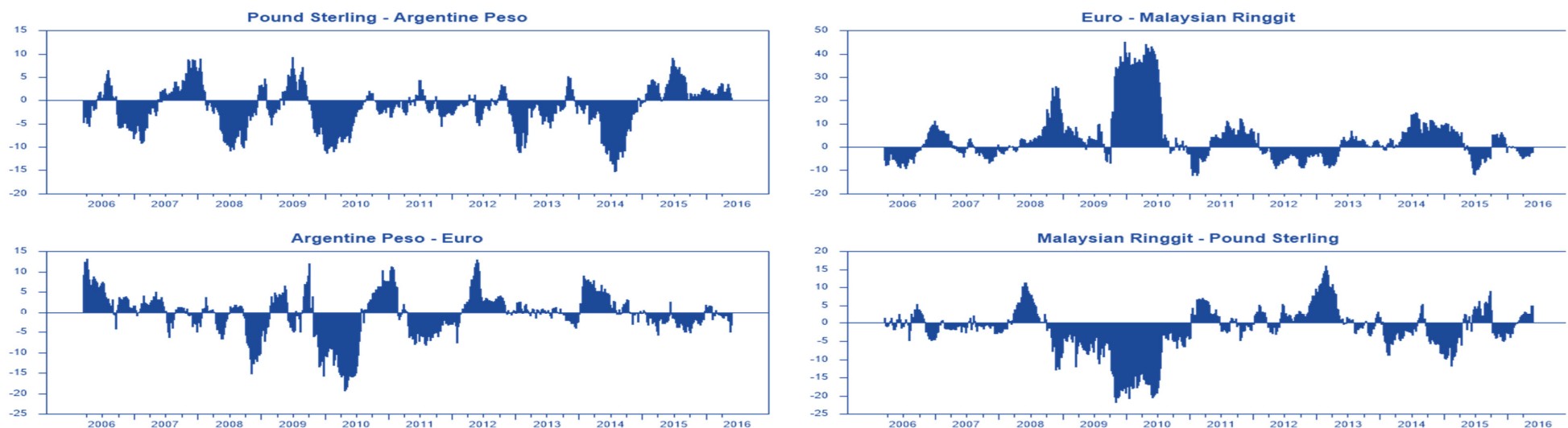

**Figure 8.** This figure shows the net volatility spillovers between developed and developing countries. In particular, the net volatility spillovers from/to the Pound Sterling (GBP), Argentine Peso (ARS), Euro (EUR), and the Malaysian Ringgit (MYR).

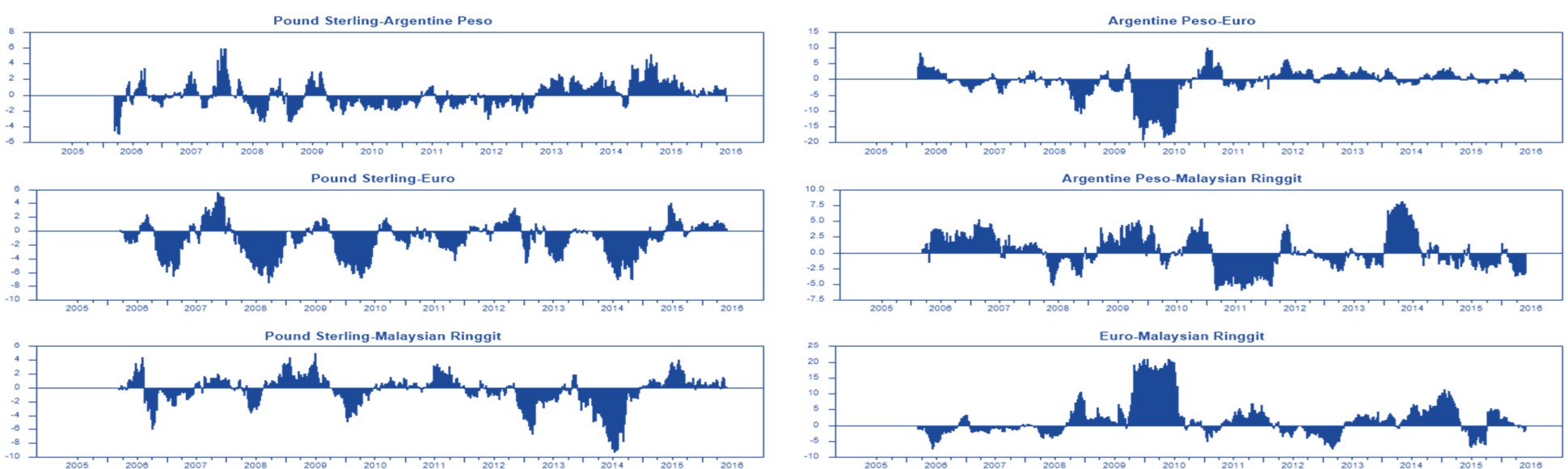

**Figure 9.** This figure shows the net pairwise volatility spillovers between developed and developing countries. In particular, the net pairwise volatility spillovers from/to the Pound Sterling (GBP), Argentine Peso (ARS), Euro (EUR), and the Malaysian Ringgit (MYR).

## 8. Conclusions

The critical question is whether the effects of return and volatility spillovers are bidirectional between developed and developing countries. Thus, in this paper, we examined the impact of return and volatility spillovers on global foreign exchange markets across developed and developing countries. Quoted against the US dollar, the data sample was comprised of twenty-three global currencies across developed and developing countries. Seven of the currencies are the most actively traded globally, including the British pound sterling (GBP), the Euro (EUR), the Australian dollar (AUD), the Swiss franc (CHF), the Icelandic krona (ISK), the Czech Republic koruna (CZK), and the Hong Kong dollar (HKD). We discussed the effect of return and volatility spillovers between developed and developing countries, using the generalised vector autoregressive (VAR) methodology. We provided the empirical results of the spillover index, in the form of a static analysis of "the spillover tables", as well as a dynamic analysis in the form of "spillover plots". We also discussed the time-varying volatility spillovers among developed and developing countries using autoregressive conditional heteroskedasticity (ARCH).

During the time period of the sample investigation (2005–2016), several exciting economic events reveal the magnitude and extent of the volatility spillovers' effect across global foreign exchange markets, in particular, from the perspective of the recent financial markets' interconnectedness. Nevertheless, the findings did not disclose evidence of bidirectional spillovers between developed and developing countries. However, we found non-negligible evidence of unidirectional spillovers (Table 4) from developed to developing countries. We also found that developed countries acted as receivers and transmitters of volatility, dominated by the British pound sterling (GBP), Australian dollar (AUD), and the Euro (EUR), whereas developing countries were net receivers of volatility.

Furthermore, the empirical results conclusively show that the magnitude and extent of the return and volatility spillovers were significantly large within the European region (Eurozone and non-Eurozone currencies), in particular, during crisis episodes, whereby the volatility spillovers replicate remarkable bursts.

This phenomenon is in line with the findings presented by Glick and Rose (1998) and Yarovaya et al. (2016) that currency crises tend to be regional. From a policy point of view, in this paper, we document significant macroeconomic implications. Firstly, the extent of global foreign exchange markets' volatility channels highlight the significance of contagion and systemic risk, particularly from the global systemically important financial institutions. Secondly, the substantial return spillovers between developed countries, especially within the European region (Eurozone and non-Eurozone currencies), further quantify the importance of cross-market linkages and the recent financial innovations. In addition, it opens avenues for a better understanding of the potential crisis of a highly interlinked nature, mirrored in the historical economic events.

Overall, this paper contributes to the literature of intra-foreign exchange markets' channels from the perspective of developed and developing countries. Here, the empirical results show that the spillover channels between developed and developing countries are insignificant. However, this raises the question of how the recent financial turmoil (which affected both developed and developing countries) propagated across the global economies.

## 9. Study Limitations and Future Research

Examining the spillover effects between developed and developing countries over the time period 2005–2016 is not a key limitation, since we were mainly interested in the time pre and post the 2008 financial crisis. Thus, in this paper, we investigated the effect of return and volatility spillovers during bad times between developed and developing countries, i.e., crisis periods. A functional area for future research is to examine the magnitude and extent of return and volatility spillovers during good times between developed

and developing countries, since, after the time period of our study (2005–2016), new exciting economic events happened. Our findings show that volatility spillovers are significantly associated with financial crises and economic events. From the viewpoint of policymakers, the high level of financial interconnectedness within the European countries is of extreme concern.

**Funding:** This research received no external funding.

**Informed Consent Statement:** Not applicable.

**Data Availability Statement:** The data presented in this study are openly available in Mendeley Data at doi:10.17632/p48ns5bzny.1.

**Conflicts of Interest:** The authors declare no conflict of interest.

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
