# Peer review of "Volatility Spillovers among Developed and Developing Countries: The Global Foreign Exchange Markets"

_jrfm, doi:10.3390/jrfm14060270_

Round 1
Reviewer 1 Report
In this paper the author analyses the volatility spillovers transmission of currency exchange rates across developed and developing countries during the period between 2005 – 2016. The methods are correct but I believe that the paper has some problems which make it not proper to be accepted for publication.
The main problem is the complete use of out of date information. What is the actual interest of having a study from 2005 to 2016? What is the added value of this paper to the literature? Which is the main advance of this paper? It uses a well stablished methodology, which is not a problem, unless it is used to new data, for example.
Telling that "Overall, this paper is the first (to our knowledge) to document the transmission of returns and volatility spillover between developed and the developing countries" is not serious in this context. Just two examples: https://www.sciencedirect.com/science/article/pii/0927538X94000297 and https://www.sciencedirect.com/science/article/pii/S0378426611001178.
Moreover, the author does not have an updated literature review. The author just presents a paper from 2017 (and even in this case it doesn't match with the reference in the main text). Probably this is related with the fact that the sample is out of date and probably the author uses an older work (for example, part of a PhD). But it is absolutely necessary to update the work.
Besides this, the author uses a VAR approach, which is not incorrect, but it should be compared with other possible methodologies.
Author also has some spelling/gramatical errors which should be cleaned. For example, author refers to the 2008th financial crisis. Certainly he/she wants to say 2008 (once there are not other 2007 known financial crises). But this is not the only error.
The author should also have caution with the way he presents some information. For example, in Figures are presented "Series of European Sovereign Debt Crisis". Firstly, if the author wants to refer more than one, should be "crises". But what is the real intention to identify several crises if the crisis was the same, although with possible different moments?
Figures 1 to 4 are totally copied from other sources. The author didn't care with it, just referencing "Figure 1" and so on, but not fulfilling the requirements of the guidelines. All figures should have a caption. This occurs also in other figures.
Some tables also have problems. For example, table 5 is presented as "Table 5: Regression (ehat2 L.ehat2)". Readers do not have to know that ehat2 and L.ehat2 are normally the names that used use to call their variables in statistical softwares. But I could call them, for example, ABCDE and FGHIJ. But in the paper they should be clearly identified in a proper way.
What to say about Table 7?
Another problem is that the paper is too long. Once again it seems that the author used a part from an older work, without filtering it. Note that it is normal to use parts of a PhD, for example, to make one or different papers (I did it, for example), but it implies some work that the author didn't care.
Finally, MDPI papers do not use footnotes. It is always important to read all the guidelines.
Author Response
Response to Reviewer 1 Comments
Dear reviewer,
Thank you very much for taking the time to assess my manuscript. I am pleased to inform you that I have managed to address all the concerns you have raised apart from the manuscript size. I have managed to redesign the manuscript and reduce its size from 44 pages to 41. I could not reduce its size further because the manuscript contains 5 tables and a very large 9 figures, which acquire a total of 13 pages.
Point 1: The main problem is the complete use of out of date information. What is the actual interest of having a study from 2005 to 2016? What is the added value of this paper to the literature? Which is the main advance of this paper? It uses a well stablished methodology, which is not a problem, unless it is used to new data, for example.
Response 1: An extra discussion involving recently published papers is added to the literature review section to bring the paper up to date. Since we investigate the spillovers effect between developed and developing countries; the aim is to provide comprehensive and precise measures of return spillover and volatility spillover pre-and-post the recent financial crisis of 2007-09. Besides the spillover index methodology; the paper also uses the autoregressive conditional heteroskedasticity (ARCH) model.
Point 2:Telling that "Overall, this paper is the first (to our knowledge) to document the transmission of returns and volatility spillover between developed and the developing countries" is not serious in this context. Just two examples: https://www.sciencedirect.com/science/article/pii/0927538X94000297 and https://www.sciencedirect.com/science/article/pii/S0378426611001178.
Response 2: The argument ‘’Overall, this paper is the first (to our knowledge) …’’ is now removed from the paper.
Point 3: Moreover, the author does not have an updated literature review. The author just presents a paper from 2017 (and even in this case it doesn't match with the reference in the main text). Probably this is related with the fact that the sample is out of date and probably the author uses an older work (for example, part of a PhD). But it is absolutely necessary to update the work.
Response 3: I have managed to update the literature review section. The paper from 2017 is now cited in the main text, and other recently published papers (2019 and 2020) are also added to the literature review.
Point 4: Besides this, the author uses a VAR approach, which is not incorrect, but it should be compared with other possible methodologies.
Response 4: This comment is also accommodated. The comparison between VAR and other methodologies is clearly stated.
Point 5: Author also has some spelling/gramatical errors which should be cleaned. For example, author refers to the 2008th financial crisis. Certainly he/she wants to say 2008 (once there are not other 2007 known financial crises). But this is not the only error.
Response 5: A great effort is made to clean all typos and grammatical errors.
Point 6: The author should also have caution with the way he presents some information. For example, in Figures are presented "Series of European Sovereign Debt Crisis". Firstly, if the author wants to refer more than one, should be "crises". But what is the real intention to identify several crises if the crisis was the same, although with possible different moments?
Response 6: This issue has been resolved now and changed to the European Sovereign Debt Crisis in 2009 instead.
Point 7: Figures 1 to 4 are totally copied from other sources. The author didn't care with it, just referencing "Figure 1" and so on, but not fulfilling the requirements of the guidelines. All figures should have a caption. This occurs also in other figures.
Response 7: A caption is added to all the figures and tables and now the paper meets all the guidelines.
Point 8: Some tables also have problems. For example, table 5 is presented as "Table 5: Regression (ehat2 L.ehat2)". Readers do not have to know that ehat2 and L.ehat2 are normally the names that used use to call their variables in statistical softwares. But I could call them, for example, ABCDE and FGHIJ. But in the paper they should be clearly identified in a proper way.
Response 8: This comment accommodated on pages 31 and 32. I have also cleaned the paper from the names of all unnecessary statistical variables.
Point 9: What to say about Table 7?
Response 9: Extra clarification is added along with a caption; table 7 shows the result of the conditional variance of the estimated Arch (1) model.
Point 10: Another problem is that the paper is too long. Once again it seems that the author used a part from an older work, without filtering it. Note that it is normal to use parts of a PhD, for example, to make one or different papers (I did it, for example), but it implies some work that the author didn't care.
Response 10: An effort is made to reduce the size of the paper. I have managed to redesign the paper and reduce its size from 44 pages to 41. However, it is difficult to reduce the size as the paper has 5 tables and large 9 figures.
Point 11: Finally, MDPI papers do not use footnotes. It is always important to read all the guidelines.
Response 11:This comment is accommodated; I have incorporated all the necessary footnotes into the main text.
Kind Regards,
Walid A Mohammed
Reviewer 2 Report
Title: Volatility Spillover between Developed and Developing Countries: The global Foreign Exchange Market’s Channel
- At first, It would be useful if the authors explain the differences of their methodology with realized volatility estimators (see for instance Floros et al. 2020).
- What will be the economic implications in macro – level in business activity?
- It would also be useful for the audience and future researchers if a guide for the future research is provided: how this research could be used concretely to open new pathways? Is it possible to provide some examples and possible directions for future research?
This is a good work and I think that a revised version with the abovementioned concerns could be a contribution to the literature.
Literature
Floros C., Gkillas K., Konstantatos C., Tsagkanos A., (2020) “Realized measures to explain volatility changes over time” Journal of Risk Financial Management. Vol 13(6), 125-144
Author Response
Dear Reviewer,
Thank you very much for taking the time to assess my manuscript and for the constructive comments. I am pleased to inform you that I have managed to address all the concerns you have raised.
Point 1: At first, It would be useful if the authors explain the differences of their methodology with realized volatility estimators (see for instance Floros et al. 2020).
Response 1: This comment is accommodated in the introduction section. The comparison between VAR; realised volatility (RV) estimator and other methodologies is clearly stated.
Point 2: What will be the economic implications in macro – level in business activity?
Response 2: An extra discussion regarding economic implication is added on page 36.
Point 3: It would also be useful for the audience and future researchers if a guide for the future research is provided: how this research could be used concretely to open new pathways? Is it possible to provide some examples and possible directions for future research?
Response 3: This comment is accommodated on page 36; further discussion regarding a future research is added.
Kind Regards,
Walid A Mohammed
Reviewer 3 Report
The authors analyze the volatility spillover between developed and developing countries in the foreign market exchange at global level.
The literature review presents sufficient information on the foreign exchange market’s spillover channel.
The methodology and data are suited for the studied topic and based on previous studies.
The results are well presented and corelated with the ones from previous studies.
The conclusions are supported by the results.
Author Response
Dear Reviewer,
I would like to take this opportunity to thank you very much for taking the time to assess my manuscript and for your constructive comments.
Kind Regards,
Walid A Mohammed
Round 2
Reviewer 1 Report
I believe (and hope) that the authors could have made the upload of the wrong file, once it doesn't fit with the answers of the authors considering my comments of the previous review. So, it will be necessary to see that new version to evaluate correctly the paper.
Author Response
Dear Reviewer,
Once again, thank you for your valuable comments, and yes. I have uploaded the wrong manuscript but this time I will make sure that the manuscript uploaded is the right one.
Round 3
Reviewer 1 Report
I consider that the made some interesting improvements in the paper. Although, I continue to feel that some issues are not sufficiently explained. In particular:
1. The problem of the out of date of the study was not treated. Authors do not clearly state at least one of two different issues: i) what is the relevance of a study like this, when it deals with data until 2016; ii) identify this as possible shortcoming, once since 2016 the spillovers could have changed and today, with the presence of another "exciting economic event" (as identified by the authors in page 30) and the information could be not relevant.
2. Authors continue statintg "this paper contributes to the scarce literature of intra-foreign exchange markets’ channel". The literature about this topic, definitely, is not scarce.
3. Some figures have captions which are not sufficiently explained. For example: figure 5 is the ARCH(1) model. Ok, but for what? Figures in pages 28 and 29 not fulfill the guidelines. The same in tables (for example, table 7 continues not to be understandable!)
Author Response
Dear Reviewer,
Thank you for taking the time continue assessing my manuscript; your comments are very helpful and highly appreciated. Now, I have managed to accommodate almost all the comments apart from the figures on pages 28-29 as there is no enough space to add more captions. Please see my responses below;
Point 1: The problem of the out of date of the study was not treated. Authors do not clearly state at least one of two different issues: i) what is the relevance of a study like this, when it deals with data until 2016; ii) identify this as possible shortcoming, once since 2016 the spillovers could have changed and today, with the presence of another "exciting economic event" (as identified by the authors in page 30) and the information could be not relevant.
Response 1: This comment is now accommodated. I have included a limitations section where I highlighted the shortcoming of the study including some possible ways to overcome such limitation in a future study.
Point 2: Authors continue stating "this paper contributes to the scarce literature of intra-foreign exchange markets’ channel". The literature about this topic, definitely, is not scarce.
Response 2: This argument is removed from the paper now.
Point 3: Some figures have captions which are not sufficiently explained. For example: figure 5 is the ARCH(1) model. Ok, but for what? Figures in pages 28 and 29 not fulfil the guidelines. The same in tables (for example, table 7 continues not to be understandable!)
Response 3: a clear caption is added to figure 5 and tables (6 and 7). However, it is very difficult to add extra captions to the figures on pages 28 – 29 as there is not space.
Round 4
Reviewer 1 Report
Some figures continue with problems in the captions, without following the guidelines. Although, in terms of contents, the paper is now in conditions to be accepted. I believe that the authors could make the relevant corrections with the editorial office.